# Hormonal contraception alters vaginal microbiota and cytokines in South African adolescents in a randomized trial

Christina Balle[1], Iyaloo N. Konstantinus[1], Shameem Z. Jaumdally[1], Enock Havyarimana [1], Katie Lennard[2], Rachel Esra[1], Shaun L. Barnabas[1,3], Anna-Ursula Happel[1], Zoe Moodie [4], Katherine Gill [3], Tanya Pidwell[3], Ulas Karaoz [5], Eoin Brodie [5,6], Venessa Maseko[7], Hoyam Gamieldien[1], Steven E. Bosinger [8], Landon Myer[9], Linda-Gail Bekker[3], Jo-Ann S. Passmore [1,10,13] & Heather B. Jaspan [1,11,12,13 ✉]

Young women in sub-Saharan Africa are disproportionally affected by HIV infection and unintended pregnancies. However, hormonal contraceptive (HC) use may influence HIV risk through changes in genital tract microbiota and inflammatory cytokines. To investigate this, 130 HIV negative adolescent females aged 15–19 years were enrolled into a substudy of UChoose, an open-label randomized crossover study (NCT02404038), comparing acceptability and contraceptive product preference as a proxy for HIV prevention delivery methods. Participants were randomized to injectable norethisterone enanthate (Net-En), combined oral contraceptives (COC) or etonorgesterol/ethinyl estradiol combined contraceptive vaginal ring (CCVR) for 16 weeks, then crossed over to another HC for 16 weeks. Cervicovaginal samples were collected at baseline, crossover and exit for characterization of the microbiota and measurement of cytokine levels; primary endpoints were cervical T cell activation, vaginal microbial diversity and cytokine concentrations. Adolescents randomized to COCs had lower vaginal microbial diversity and relative abundance of HIV risk-associated taxa compared to Net-En or CCVR. Cervicovaginal inflammatory cytokine concentrations were significantly higher in adolescents randomized to CCVR compared to COC and Net-En. This suggests that COC use may induce an optimal vaginal ecosystem by decreasing bacterial diversity and inflammatory taxa, while CCVR use is associated with genital inflammation.

[1] Department of Pathology, Institute of Infectious Disease and Molecular Medicine, University of Cape Town, Cape Town, South Africa. [2] Department of Integrative Biomedical Sciences, University of Cape Town, Cape Town, South Africa. [3] Desmond Tutu HIV Centre, University of Cape Town, Cape Town, South Africa. [4] Vaccine and Infectious Disease Division, Fred Hutchinson Cancer Research Center, Seattle, WA, USA. [5] Earth and Environmental Science, Lawrence Berkeley National Laboratories, Berkeley, CA 94720, USA. [6] University of California, Berkeley, CA, USA. [7] National Institute for Communicable Diseases, Sandringham, Johannesburg, South Africa. [8] Department of Pathology & Laboratory Medicine, Emory University School of Medicine; Division of Microbiology and Immunology, Yerkes National Primate Research Center, Atlanta, GA, USA. [9] Division of Epidemiology, Biostatistics, School of Public Health & Family Medicine, University of Cape Town, Cape Town, South Africa. [10] National Health Laboratory Service, Cape Town, South Africa. [11] Seattle Children's Research Institute, Seattle, WA, USA. [12] University of Washington Department of Pediatrics and Global Health, Seattle, WA, USA. [13] These authors contributed equally: Jo-Ann S. Passmore, Heather B. Jaspan. ✉email: hbjaspan@gmail.com

A dolescent females and young women are at high risk of sexually transmitted infections (STI), particularly in sub-Saharan Africa (SSA)[1,2]. Young women in SSA are also faced with a high risk of unintended pregnancies[3,4], which can be associated with increased maternal and infant mortality and morbidity, especially in developing countries[5]. Hormonal contraceptives (HC) play a crucial role in preventing unintended pregnancy. However, a number of observational studies have suggested that the use of HCs may influence women's susceptibility to HIV infection[6,7], by altering vaginal microbiota[8–15] or immunity[16–21].

The composition of the vaginal microbiota may impact HIV susceptibility[8,9]. Highly diverse communities, as well as the relative and absolute abundance of specific organisms, including *Parvimonas*, *Prevotella* spp. (such as *P. amnii*, *P. bivia*, and *P. melaninogenica*), *Mycoplasma*, *Gemella asaccharolytica*, *Sneathia*, *Veillonella montpellierensis*, and *Megasphaera*, have been associated with HIV risk[9,10]. In contrast, lactobacilli, particularly non-*iners Lactobacillus* spp., have been associated with lower risk of HIV infection[9–11]. High-diversity vaginal communities are also strongly associated with high levels of genital pro-inflammatory cytokine concentrations[20,21], which increase the risk for subsequent HIV infection[19]. In observational cohorts, HCs have been associated with cervical and vaginal inflammation[16,17]. Combined oral contraceptive (COC) use has been associated with increases in relative abundance of lactobacilli[12]. There are conflicting data as to whether HCs are protective or permissive for bacterial vaginosis (BV)[13–15,18].

Estrogen is associated with glycogen deposition in epithelial cells, aiding lactic acid metabolism[22,23]. The progestin-only injectable depomedroxyprogesterone acetate (DMPA) may cause a hypoestrogenic state, which in turn may impair colonization of lactic acid-producing lactobacilli[24]. In observational studies, DMPA has been associated with decreased risk of BV; however, there are few data on the effects of the injectable norethisterone enanthate (Net-En) or combined contraceptive vaginal rings (CCVR)[13,15]. We hypothesized that the long-acting injectable progestin Net-En may alter vaginal microbiota, causing increased genital inflammation, but that COC and CCVR, due to their estrogen component, would not. Therefore, we assessed temporal changes in cervicovaginal microbiota and cytokines in adolescents randomized to three HCs: the long-acting progestin-only injectable Net-En, combined oral contraceptives (COC) Triphasil® or Nordette® and NuvaRing®, a CCVR containing etonorgesterol/ethinyl estradiol. We found that adolescents assigned to COC not only had more lactobacilli compared to those assigned to Net-En, but also those assigned to CCVR. CCVR use was associated with significantly elevated genital tract inflammation compared with both Net-En and COC. Together, these data suggest that COC may benefit women at high risk for BV and HIV.

## Results

**Cohort characteristics**. Of the 180 screened for the parent study, 130 adolescent females (median age 17 years; interquartile range (IQR) 16–18) were enrolled into a randomized study with crossover (Fig. 1)[25]. Of these, 45 were randomized to NuvaRing® (hereon referred to as CCVR), 45 to Net-En, and 40 to COC (Table 1 and Fig. 1). All particiants consented to enrol in this substudy, which intended to assess mucosal changes induced by contraceptives. A total of 108 adolescents reached the crossover visit at 16 weeks, while 94 adolescents completed the 32-week exit visit. One participant assigned to the Net-En arm seroconverted before the crossover visit, one participant withdrew consent for mucosal sampling at the exit visit, and there was one pregnancy

in the CCVR arm before the crossover visit; therefore, these three participants were exited from the substudy. At baseline, adolescents assigned to each study arm were similar in demographics, medical, and reproductive history (including age, body mass index (BMI), vaginal insertion practices and antibiotics use, as well as BV, *Candida*, and STI prevalence) (Table 1). Reported sexual risk behavior was similar across groups, as was the distribution of baseline HC use.

**Vaginal microbiota in South African adolescents**. Of the 329 vaginal lateral wall samples taken from 130 adolescents at all visits, 319 samples passed sequencing and quality control measures (≥5000 reads/sample, screening: $n = 126$, crossover: $n = 104$, and exit: $n = 89$). Using soft k-means clustering[26] with weighted UniFrac as the distance measure, we identified three major microbial community-state types (CSTs) (corresponding to CST-I, CST-III, and CST-IV described by Ravel et al.[27], Fig. 2a and Supplementary Fig. 1). The most common community type, CST-IV ($n = 145$, 45.5%), had high microbial diversity comprising a diverse group of anaerobic bacteria with the most abundant species being *Gardnerella vaginalis* followed by *Lachnovaginosum* genomospecies (BVAB1), *Megasphaera*, *L. iners*, *Prevotella* spp. (including *P. amnii*, *P. timonensis*, and *P. bivia*), *Atopobium vaginae*, *Sneathia*, and *Aerococcus christensenii* (Fig. 2a). The latter two communities, CST-I and CST-III, were low-diversity communities dominated by *L. crispatus* ($n = 76$, 23.8%) and *L. iners* ($n = 98$, 30.7%), respectively (Fig. 2a and Supplementary Fig. 1). The majority of adolescents with a CST-IV community were correspondingly BV positive (Nugent scores 7–10; $n = 126$, 86.9%), whereas adolescents with CST-I and CST-III communities were mostly BV negative (Nugent 0–3; $n = 73$, 96.1% and $n = 84$, 85.7%, respectively). In comparison to studies performed in North American cohorts[27,28], the vaginal community types CST-II (dominated by *L. gasseri*) and CST-V (dominated by *L. jensenii*) were not present in this cohort, similar to what has been found in other African cohorts[20,21]. Baseline alpha diversity (i.e., within-sample diversity) was slightly higher in adolescents who were randomized to the Net-En arm compared to those assigned to COC and CCVR (median Shannon Index (SI) 1.78 (IQR 0.64–2.27) vs. 0.87 (0.43–1.99) and 1.46 (0.61–2.14), respectively), but not significantly so (Table 1). Importantly, CSTs according to the randomized arm at baseline were evenly distributed.

**Impact of HC on the vaginal microbiota**. The impact of HC after 16 weeks of contraceptive use was first assessed in an intention-to-treat (ITT) analysis. There were no differences in reported sexual risk behavior between arms at crossover. Furthermore, no differences in the presence of yeast (by microscopy), BV (by Nugent scoring), or in HSV-2 serostatus were observed between study arms at 16 weeks (Table 2).

At crossover, after 16 weeks of randomized contraceptive use, a significant difference in the distribution of vaginal CSTs between randomized arms was observed ($P = 0.029$) (Table 2). This significant difference was driven by the difference between Net-En and COC ($P = 0.007$). There was some evidence of a difference between Net-En and CCVR ($P = 0.080$) but no evidence of a difference between COC and CCVR ($P = 0.479$). The CST-III community type was the most prevalent in the COC arm and significantly more prevalent in this arm compared to the Net-En study arm (adj. $P = 0.019$, Table 2). In both the Net-En and the CCVR arms, the diverse CST-IV community type was most common followed by the *L. crispatus*-dominant CST-I community type in the Net-En arm and the *L. iners*-dominant CST-III community type in the CCVR arm. This corresponded to a high proportion of participants in the COC arm transitioning

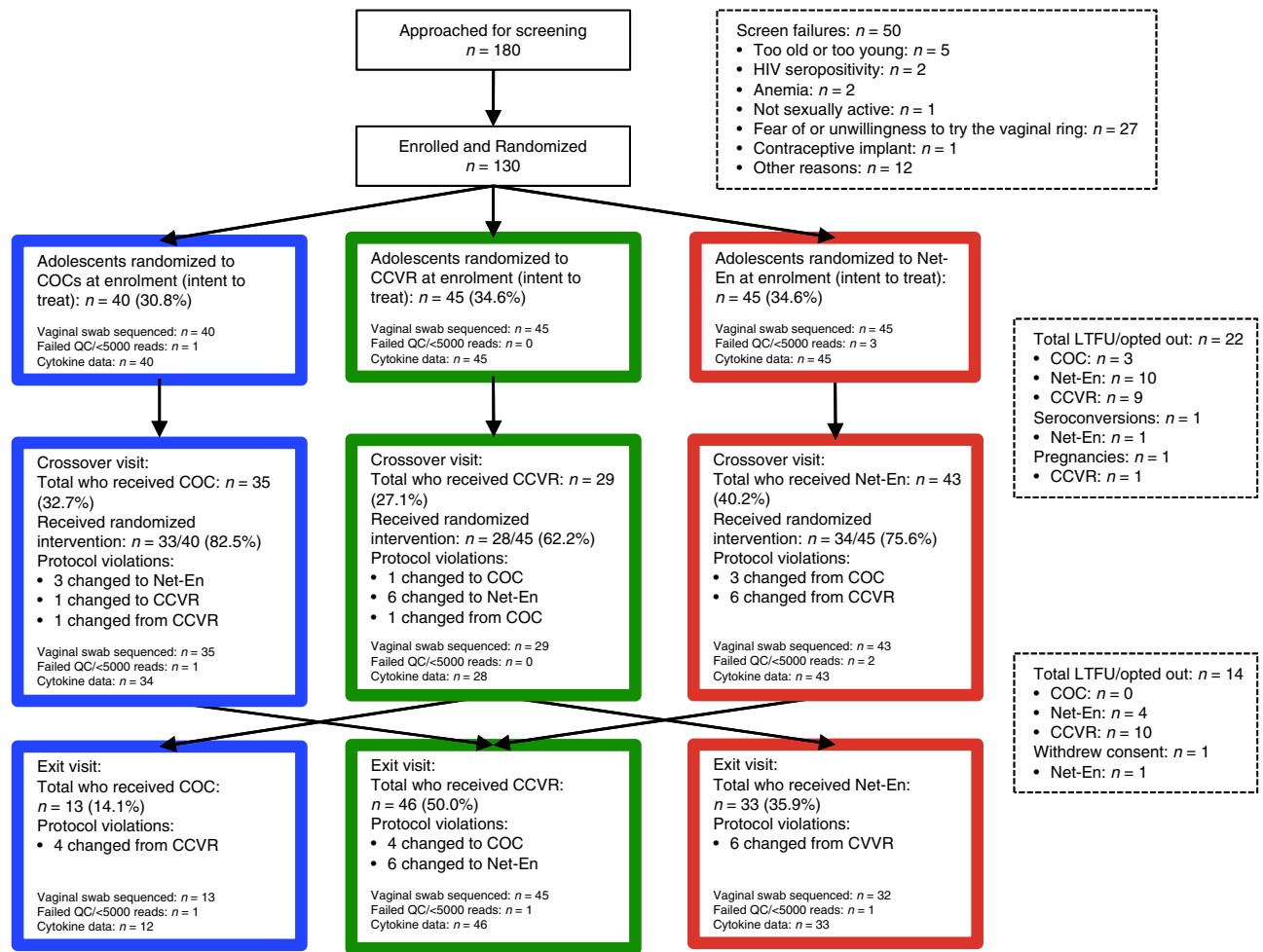

**Fig. 1 Study overview and randomization.** CONSORT diagram of the number of participants that completed each study visit and provided genital samples, and the distribution of participants on each of the hormonal contraceptive methods analyzed: combined oral contraceptives (COC), the Net-En injection, and the vaginally inserted combined contraceptive ring (CCVR) at crossover and exit. Note that at crossover, some participants could choose their next contraceptive method if assigned to CCVR as their first method. LTFU loss-to-follow-up.

from CST-I or CST-IV to CST-III (43% and 35%, respectively), while the majority of participants with CST-III at baseline remained stable (67%) (Fig. 2b–d; Supplementary Table 1). In the CCVR arm, 67% of participants with CST-IV at baseline stayed in CST-IV, while 30% and 33% of participants with CST-I or CST-III respectively transitioned to CST-IV. Participants in the Net-En arm remained stable with no transitions away from CST-I and with 72% of participants with CST-IV at baseline not transitioning. However, none of the participants in the Net-En arm with a CST-III community at baseline stayed as such: 57% transitioned to CST-I and 43% transitioned to CST-IV. Using omnibus symmetry exact tests, there was no strong statistical evidence for a lack of symmetry in the CSTs at baseline and at crossover apart from some evidence of a trend for the Net-En group ($P = 0.067$) where all those with CST-III at baseline transitioned to another state at crossover.

A decrease in alpha diversity was observed in the COC and Net-En arms from baseline to crossover, while an increase was observed in the CCVR arm; however, this did not reach statistical significance (Fig. 2e–g and Table 2). Cross-sectionally, the vaginal microbiota of adolescents randomized to the COC arm had significantly lower alpha diversity at crossover (median SI: 0.90 (IQR 0.35–1.54)) compared to the microbiota of adolescents assigned to either the Net-En (median SI: 1.64 (IQR 0.75–2.22))

or CCVR (median SI: 1.66 (IQR 0.79–1.95), adj. $P = 0.009$ and $P = 0.021$, respectively, Table 2). No difference in vaginal alpha diversity was evident at crossover between adolescents randomized to the CCVR and Net-En arms (adj. $P = 0.941$). Using a linear regression model with Tukey's post hoc test, the alpha diversity of the vaginal microbiota of adolescents in the COC arm was significantly lower than that of adolescents assigned to the CCVR arm ($P = 0.018$) and the Net-En arm ($P = 0.007$). After adjusting for possible confounders (antibiotic use (collinear with baseline STIs)), these differences remained significant (COC vs. CCVR, $P = 0.007$ and COC vs. Net-En, $P = 0.003$). We assessed for effect of modification by baseline CST by including interaction terms for study arm and baseline CST in the model; however, these were not significant ($P = 0.401$), suggesting no evidence that the effect of study arm on vaginal alpha diversity at crossover differs by baseline CST. The contribution of potential mediators, such as menstrual bleeding patterns and sexual risk behavior, of the effects of study arm on the microbiota, was assessed using a mediation analysis of time since the last menstrual period and condom use, respectively (Supplementary Table 2). Only a very small and nonsignificant proportion of the effects of study arm on the microbiota could be explained by the indirect effects of menstrual cycle and sexual risk behavior, with estimates of the proportion mediated ranging from 0.0004 for the

**Table 1 Characteristics of participants at baseline according to randomization arms.**

| | COC (n = 40) (30.8%) | Net-En (n = 45) (34.6%) | CCVR (n = 45) (34.6%) |
|---|---|---|---|
| Age at screening, median yrs (IQR) | 17 (16–18) | 17 (16–18) | 17 (16–18) |
| BMI, median (IQR) | 25.5 (21.5–28.1) | 25.0 (22.7–29.7) | 24.9 (21.9–27.7) |
| *STI prevalence* | | | |
| Any STI(s) | 15 (37.5%) | 21 (46.7%) | 19 (42.2%) |
| Ct | 12 (30.0%) | 17 (37.8%) | 14 (31.1%) |
| Ng | 3 (7.50%) | 5 (11.1%) | 5 (11.1%) |
| Tv | 4 (10.0%) | 4 (8.89%) | 4 (8.89%) |
| Mg | 1 (2.50%) | 2 (4.44%) | 0 (0.00%) |
| *BV prevalence* | | | |
| BV positive | 16 (40.0%) | 22 (48.8%) | 19 (42.2%) |
| BV intermediate | 4 (10.0%) | 5 (11.1%) | 3 (6.67%) |
| BV negative | 20 (50.0%) | 18 (40.0%) | 23 (51.1%) |
| *CST distribution*[a] | | | |
| CST-I | 9 (22.5%) | 8 (19.0%) | 11 (25.0%) |
| CST-III | 13 (32.5%) | 11 (26.2%) | 10 (22.7%) |
| CST-IV | 17 (45.0%) | 23 (54.8%) | 24 (52.3%) |
| *Vaginal pH, mean (sd)* | 4.8 (4.3–5.4) | 5.0 (4.3–5.6) | 4.8 (4.3–5.4) |
| >4.5, n (%) | 29 (72.5%) | 35 (77.8%) | (71.1%) |
| Shannon index, median (IQR)[a] | 0.87 (0.43–1.99) | 1.78 (0.64–2.27) | 1.46 (0.61–2.14) |
| HSV-2 serology[b] | 14 (35.0%) | 13 (28.9%) | 12 (26.7%) |
| Yeast cells present | 6 (15.0%) | 4 (8.89%) | 10 (22.2%) |
| *Inflammation category*[c] | | | |
| High | 20 (51.3%) | 23 (56.1%) | 28 (62.2%) |
| Low | 19 (48.7%) | 18 (43.9%) | 17 (37.8%) |
| Days since the last menstrual period, median (IQR)[d] | 41 (20–229) | 42 (19–105) | 37 (14–123) |
| Antibiotic use (past 3 months) | 0 (0.00%) | 4 (8.9%) | 4 (8.89%) |
| Age menarche, median (IQR)[e] | 13 (12–14) | 13 (12–14) | 13 (12–14) |
| Tanner, median (IQR)[f] | 4.0 (4.0–4.0) | 4.0 (4.0–4.0) | 4 .0 (4.0–4.0) |
| *Parity*[g] | | | |
| Previously pregnant | 4 (10.3%) | 7 (15.6%) | 6 (13.3%) |
| *Use of hormonal contraception*[h] | | | |
| Naive | 1 (2.5%) | 2 (4.44%) | 2 (4.44%) |
| Not currently | 10 (25.0%) | 10 (22.2%) | 6 (13.3%) |
| Net-En | 20 (50.0%) | 21 (46.7%) | 28 (62.2%) |
| COC | 1 (2.5%) | 2 (4.44%) | 3 (6.67%) |
| DMPA | 7 (17.5%) | 7 (15.6%) | 5 (11.1%) |
| Implanon | 0 (0.0%) | 2 (4.44%) | 1 (2.22%) |
| *Intravaginal practices*[i] | | | |
| Douching | 0 (0.0%) | 1 (2.2%) | 0 (0.0%) |
| Washing with water | 5 (12.8%) | 6 (13.3%) | 5 (11.1%) |
| Washing with soap | 2 (5.13%) | 6 (13.3%) | 4 (8.89%) |
| Tampon use | 1 (2.56%) | 5 (11.1%) | 2 (4.65%) |
| Put anything else inside the vagina (medication/herbs) | 1 (2.56%) | 4 (8.89%) | 2 (4.65%) |
| *Sexual risk behavior*[g] | | | |
| Age of sexual debut, median (IQR) | 15 (14–16) | 15 (14–16) | 15 (14–16) |
| Any sexual partner(s) past year, n (%) | 37 (92.5%) | 40 (93.0%) | 42 (93.3%) |
| Multiple sexual partners past year, n (%) | 3 (7.50%) | 5 (11.6%) | 4 (8.89%) |
| New partner past year, n (%) | 9 (22.5%) | 15 (34.9%) | 12 (26.7%) |
| *General condom use* | 3 (7.50%) | 6 (14.0%) | 3 (6.67%) |
| Never | 4 (10.5%) | 3 (6.98%) | 6 (13.3%) |
| Almost never | 3 (7.50%) | 7 (16.3%) | 5 (11.1%) |
| Not sure | 19 (47.5%) | 13 (30.2%) | 14 (31.1%) |
| Almost always | 11 (27.5%) | 14 (32.6%) | 17 (37.8%) |
| Always | 25 (62.5%) | 26 (60.5%) | 27 (60.0%) |
| Condom use during last PV intercourse | 19 (47.5%) | 15 (34.9%) | 16 (35.6%) |
| *Sex with older partner (≥5 years)* | | | |
| No | 13 (32.5%) | 18 (41.9%) | 21 (46.7%) |
| Unsure | 8 (20.0%) | 10 (23.3%) | 8 (17.8%) |
| Yes | 0 (0.00%) | 1 (2.33%) | 0 (0.00%) |
| Transactional sex penile–anal intercourse | 0 (0.00%) | 0 (0.00%) | 4 (8.89%) |
| *Education*[j] | | | |
| School attendance | 36 (90.0%) | 39 (86.7%) | 37 (84.1%) |
| Highest grade, median (IQR) | 10 (8–11) | 10 (9–11) | 9 (8–10) |
| Tertiary attendance | 2 (5.00%) | 0 (0.00%) | 2 (4.55%) |

BMI: body mass index, BV: bacterial vaginosis, CCVR: combined contraceptive vaginal ring, COC: combined oral contraceptives, CST: community-state type, Ct: *Chlamydia trachomatis*, HSV-2: herpes simplex virus type-2 seropositive, IQR: interquartile range, LH: luteinizing hormone, Mg: *Mycoplasma genitalium*, Ng: *Neisseria gonorrhoea*, PV: penile–vaginal, sd: standard deviation, STI: sexually transmitted infection, Tv: *Trichomonas vaginalis*, yrs: years.
[a]Based on samples with available microbiome data (COC, n = 39; Net-En, n = 42; CCVR, n = 45).
[b]One equivocal result (CCVR, n = 1).
[c]Based on samples with available microbiome and cytokine data (COC, n = 39; Net-En, n = 41; CCVR, n = 45).
[d]Missing data from two adolescents (COC, n = 2; Net-En, n = 0; CCVR, n = 0).
[e]Missing data from five adolescents (COC, n = 4; Net-En, n = 1; CCVR, n = 0).
[f]Missing data from one adolescent (COC, n = 0; Net-En, n = 1; CCVR, n = 0).
[g]Missing data from two adolescents (Net-En, n = 2).
[h]Missing data from three adolescents (COC, n = 0; Net-En, n = 2; CCVR, n = 1).
[i]Missing data from one adolescent (COC, n = 1).
[j]Missing data from one adolescent (CCVR, n = 1).

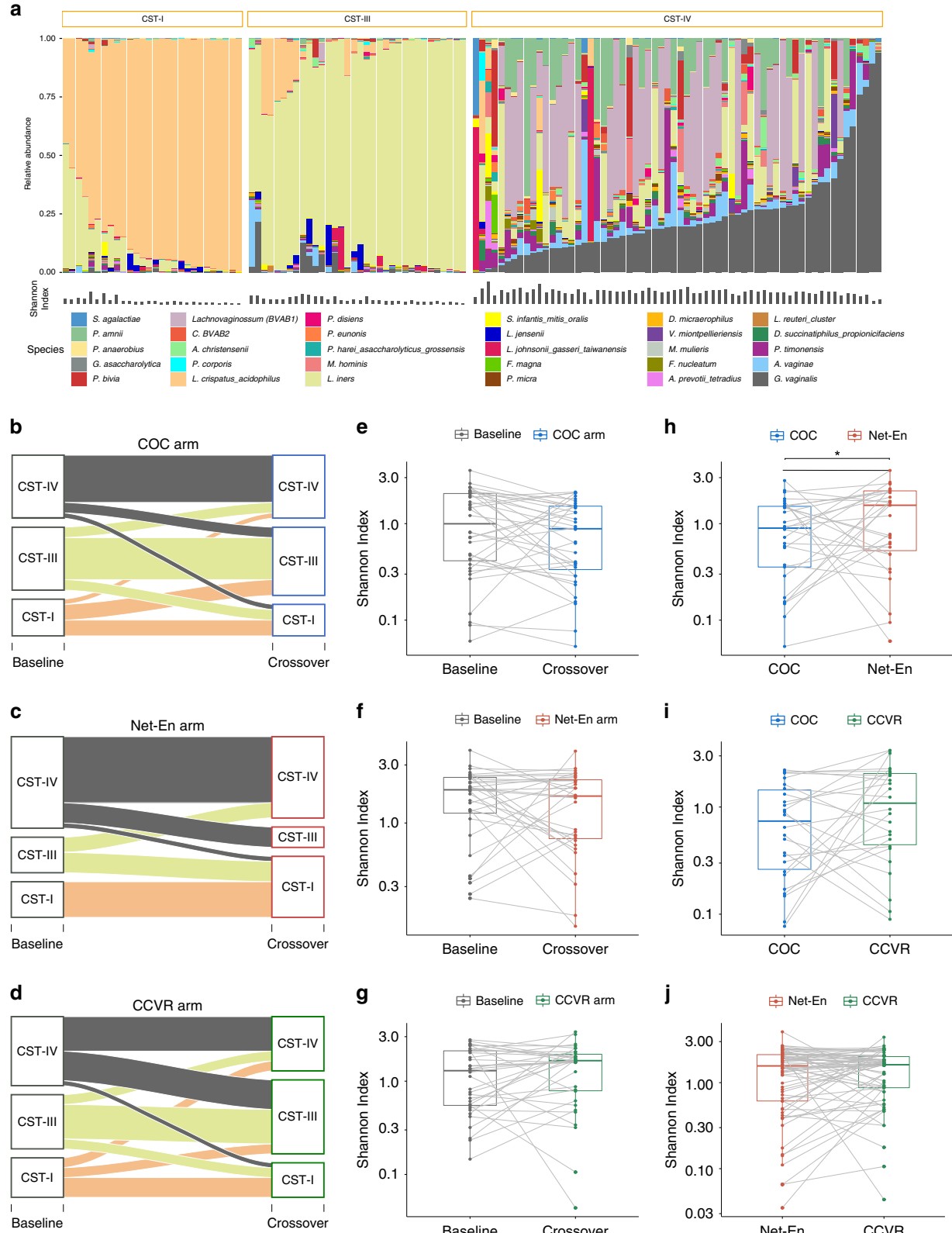

time since the last menstrual period in the COC versus CCVR arm to 0.0371 in the Net-En versus CCVR arm (Supplementary Table 2).

To assess differences in community composition cross-sectionally, we performed a permutational multivariate analysis of variance (PERMANOVA) on phylogenetic distances at crossover. We found significant differences in beta diversity between the three assigned study arms at crossover ($P = 0.026$, $R^2 = 0.040$). Pairwise comparisons revealed that this difference was driven primarily by the differences between COC and Net-En ($P = 0.008$, $R^2 = 0.048$) and to a lesser degree CCVR and COC ($P = 0.055$, $R^2 = 0.031$), while there were no significant differences between the Net-En and CCVR arm ($P = 0.383$, $R^2 = 0.012$). After adjusting for baseline CST, the results from the

**Fig. 2 Vaginal community clusters in a South African adolescent cohort. a** Barplot depicting the relative abundance of the most abundant bacteria in the baseline samples identified by 16S rRNA microbiome profiling. Samples ($n = 126$) are grouped by community-state type (CST) established using soft k-means clustering with weighted UniFrac distances and ordered by the most abundant species in each CST (CST-I: *L. crispatus*, CST-III: *L. iners*, and CST-IV: *G. vaginalis*). Alpha diversity for each sample (Shannon Index) is depicted below the barplot. **b–d** Change in CST from baseline to crossover within study arms. Alluvial plot showing the change in CST distribution from baseline to crossover for matched participants randomized to one of the three study arms: **b** COC ($n = 36$), **c** Net-En ($n = 32$), and **d** CCVR ($n = 34$) in an intention-to-treat (ITT) analysis. Each line represents one adolescent and CST changes over time. The color of the line is based on the CST assigned at baseline. **e–g** Alpha diversity at baseline and crossover within study arms. Boxplot representing in alpha diversity (Shannon Index) of vaginal microbiota for matched participants at baseline and crossover randomized to one of the three study arms: **e** COC ($n = 36$), **f** Net-En ($n = 32$), and **g** CCVR ($n = 34$). **h–j** Alpha diversity within participants changing between hormonal contraceptive methods. Boxplots showing the alpha diveriisy (Shannon Index) in **h** 62 vaginal samples from 29 participants changing from COC to Net-En or vice versa, **i** 52 vaginal samples from 26 participants changing from COC to CCVR or vice versa, and **j** 130 vaginal samples from 50 participants changing from Net-En to CCVR or vice versa. *P* values were generated using two-sided paired Wilcoxon signed-rank tests. *Y* axes are log10-transformed. Significance codes: *$P <$ 0.05. Bounds of boxes show interquartile range (IQR) with the lower and upper hinges corresponding to the 25th and 75th percentiles, respectively, lines in the middle of the box indicate median, and top and bottom whiskers demonstrate value ranges within 1.5 × IQR from the hinge. Points beyond that are plotted individually. CCVR: combined contraceptive vaginal ring, COC: combined oral contraceptives. Source data are provided as a Source Data file.

model remained similar, and the interaction of study arm and baseline CST was not significant ($P = 0.443$), indicating that there was no effect modification by baseline CST. The differences in beta diversity between the study arms at crossover remained significantly different after adjusting for antibiotic use ($P = 0.027$, $R^2 = 0.040$) again driven by differences between the COC and Net-En arm ($P = 0.011$, $R^2 = 0.048$) and the COC and CCVR arm ($R^2 = 0.031$, $P = 0.061$, Net-En vs. CCVR, $R^2 = 0.012$, $P = 0.395$).

At the crossover visit, all adolescents who were assigned to either COC or Net-En at baseline were crossed over to CCVR for the final 16 weeks, whereas adolescents assigned to CCVR at baseline were given the choice between COCs or Net-En. Some adolescents changed HC method before mucosal sampling visits due to side effects (bleeding or spotting) or issues with adherence or discomfort with study product (COC: 4/46 (8.7%); Net-En: 0/60 (0.0%); CCVR: 17/80 (21.3%), Fig. 1). Therefore, a per-protocol (PP) analysis was applied to adolescents at crossover ($n = 107$) and exit ($n = 92$). Importantly, adolescents using Net-En and CCVR reported less consistent condom use than adolescents on COC, albeit not significantly so (Supplementary Table 3). In this analysis, the vaginal microbiota of adolescents using Net-En or CCVR at the time of sampling was most commonly assigned to the diverse CST-IV community type (43.5% and 45.5%, respectively), whereas participants on COC were fairly equally distributed between CSTs (Supplementary Table 3). Using a linear mixed-effect regression model with Tukey's post hoc test, no difference in alpha diversity was observed between CCVR and Net-En users (adj. $P = 0.980$), while the vaginal microbiota of adolescents using COC had a significantly lower alpha diversity compared to the microbiota of those using Net-En or CCVR (adj. $P = 0.004$ and 0.002, respectively) (Supplementary Table 3). After adjusting for antibiotic use and baseline CST, the alpha diversity remained significantly different between COC users and both Net-En (adj. $P = 0.012$) and CCVR users (adj. $P = 0.009$). Once again, the interaction between baseline CST and HC method was non-significant ($P = 0.772$). There was some evidence of differences in microbiota composition (beta diversity) between adolescents on the three different contraceptive methods in the PP analysis; however, these did not reach statistical significance (PERMA-NOVA $P = 0.057$, $R^2 = 0.034$).

In a paired analysis of matched samples from 29 adolescents who either changed from COC use to Net-En use or vice versa, the alpha diversity of the vaginal microbiota of adolescents when on COC was significantly lower than when on Net-En (median SI: 0.89 (IQR 0.34–1.52) vs. 1.60 (0.55–2.18), $P = 0.028$, Fig. 2h). Vaginal alpha diversity of adolescents when on COC was

moderately lower than when using CCVR but not significantly so ($n = 26$, median SI: 0.75 (IQR 0.26–1.44) vs. 1.10 [0.45–2.05], $P = 0.162$) (Fig. 2i). In contrast, alpha diversity of the vaginal microbiota was similar within participants when using Net-En and CCVR ($n = 50$, median SI: 1.57 (IQR 0.61–2.11) vs. 1.62 (0.87–2.01), $P = 0.643$) (Fig. 2j).

**Differentially abundant taxa by study arm.** We used DESeq2 to identify differentially abundant taxa between (cross-sectionally) and within (pre- vs. post longitudinally) randomized study arms. Fifteen taxa were differentially abundant between vaginal specimens from adolescents in the COC versus Net-En study arms at crossover after adjusting for antibiotic use and sexual risk behavior (condom use) (Table 3). Some *Lactobacillus* spp. had higher relative abundance in the COC study arm compared with the other two arms, whereas several *Prevotella* spp. and *Mageeibacillus indolicus* (BVAB3) were more abundant in the Net-En arm and *Prevotella amnii* was more abundant in the CCVR arm compared to COC. *Prevotella disiens*, *Gemella asaccharolytica*, *Mycoplasma hominis*, and *L. jensenii* were more abundant in the CCVR arm compared to the Net-En arm. When comparing changes in relative abundance longitudinally between baseline and crossover in matched samples from the same participant within each randomized arm using DESeq2, we observed a decrease in the abundance of two *Clostridium* spp. and an increase in *L. iners* abundances between baseline and crossover within adolescents who initiated COC (Table 3). For adolescents in the Net-En arm, an increase in the abundances of *P. disiens*, *M. hominis*, and *Campylobacter* was observed after Net-En initiation (Table 3), while no taxa differed from baseline to crossover in the CCVR arm.

For bacteria that have previously been associated with high or low HIV risk[9,10], we compared read counts standardized to the median read depth in a paired analysis of participants changing between contraceptive methods longitudinally (Supplementary Fig. 2). The read counts of *L. iners* were significantly higher in participants when using COC than when using Net-En ($P = 0.031$, adj. $P = 0140$), while *Prevotella* was higher in the adolescents when on Net-En compared to COC ($P = 0.032$, adj. $P = 0140$, Supplementary Fig. 2a). *Mycoplasma* ($P = 0.019$, adj. $P = 0.086$), *Parvimonas* ($P = 0.017$, adj. $P = 0.086$), and *Sneathia* ($P = 0.020$, adj. $P = 0.086$) were significantly more abundant in adolescents when on CCVR compared to when using COC (Supplementary Fig. 2b). The read counts of *L. iners* and *P. bivia* were significantly higher in adolescents when using CCVR than when using Net-En (Supplementary Fig. 2c).

**Table 2 Characteristics of participants with microbiome data at crossover according to assigned study arm (intention-to-treat).**

|  | COC ($n = 37$) | Net-En ($n = 33$) | CCVR ($n = 34$) | P value |
|---|---|---|---|---|
| *CST distribution*[a] |  |  |  | 0.029 |
| CST-I | 7 (18.9%) | 13 (39.4%) | 7 (20.6%) |  |
| CST-III | 17 (45.9%) | 4 (12.1%) | 11 (32.4%) |  |
| CST-IV | 13 (35.1%) | 16 (48.5%) | 16 (47.1%) |  |
| *Vaginal pH, mean (sd)* | 4.7 (4.2–5.1) | 4.8 (4.2–5.5) | 5.0 (4.5–5.5) | 0.101 |
| >4.5 | 21 (56.8%) | 22 (66.7%) | 25 (73.5%) | 0.327 |
| Shannon Index (SI), median (IQR) | 0.90 (0.35–1.54) | 1.64 (0.75–2.22) | 1.66 (0.79–1.95) | 0.004 |
| Median change in SI from baseline (IQR) | −0.27 (−0.90–0.25) | −0.08 (−1.04–0.46) | 0.14 (−0.33–0.70) | 0.130 |
| HSV-2 serology | 13 (35.1%) | 10 (30.3%) | 13 (38.2%) | 0.790 |
| Yeast cells present | 4 (10.8%) | 6 (18.2%) | 6 (17.6%) | 0.629 |
| *BV prevalence* |  |  |  | 0.522 |
| BV positive | 13 (35.1%) | 16 (48.5%) | 14 (41.2%) |  |
| BV intermediate | 3 (8.11%) | 0 (0.0%) | 1 (2.94%) |  |
| BV negative | 21 (56.8%) | 17 (51.5%) | 19 (55.9%) |  |
| *STI incidence* |  |  |  |  |
| Any STI(s) | 7 (18.9%) | 9 (27.3%) | 13 (38.2%) | 0.192 |
| Ct | 4 (10.8%) | 4 (11.8%) | 8 (23.5%) | 0.273 |
| Ng | 2 (5.41%) | 1 (2.94%) | 7 (20.6%) | 0.057 |
| Tv | 1 (2.70%) | 1 (2.94%) | 1 (2.94%) | 1.000 |
| Mg | 1 (2.70%) | 3 (9.38%) | 0 (0.0%) | 0.161 |
| *Inflammation category*[a] |  |  |  | 0.149 |
| High | 16 (44.4%) | 16 (48.5%) | 22 (66.7%) |  |
| Low | 20 (55.6%) | 17 (51.5%) | 11 (33.3%) |  |
| Antibiotic use since the last visit | 16 (45.7%) | 14 (42.4%) | 10 (29.4%) | 0.416 |
| Days since the last menstrual period, median (IQR)[b] | 21 (12–78) | 51 (15–157) | 27 (13–70) | 0.354 |
| *Sexual risk behavior since the last visit*[c] |  |  |  |  |
| Any sexual partners, n (%) | 32 (91.4%) | 25 (89.3%) | 29 (90.6%) | 1.000 |
| Multiple sexual partners, n | 1 (2.86%) | 1 (3.57%) | 0 (0.00%) | 0.749 |
| New partner, n | 2 (5.71%) | 1 (3.7%) | 0 (0.0%) | 0.503 |
| Sex acts per week, median (IQR) | 1 (1–2) | 1 (1–2) | 1 (1–2) | 0.260 |
| *Condom use* |  |  |  | 0.260 |
| Never | 6 (17.1%) | 7 (25.0%) | 8 (25.0%) |  |
| Less than half the time | 6 (17.1%) | 1 (3.57%) | 3 (9.38%) |  |
| Half the time | 16 (45.7%) | 12 (42.9%) | 7 (21.9%) |  |
| More than half the time | 3 (8.6%) | 2 (7.14%) | 4 (12.5%) |  |
| Always | 4 (11.4%) | 6 (21.4%) | 10 (31.3%) |  |
| *Condom use during the last PV intercourse* |  |  |  | 0.407 |
| Yes | 21 (60.0%) | 13 (46.4%) | 20 (62.5%) |  |
| *Intergenerational sex with the older partner (≥5 years)* |  |  |  | 0.103 |
| Yes | 1 (2.86%) | 3 (10.7%) | 0 (0.00%) |  |
| Unsure | 0 (0.00%) | 0 (0.00%) | 0 (0.00%) |  |
| No | 34 (97.1%) | 24 (89.3%) | 32 (100%) |  |
| Transactional sex | 0 (0.00%) | 0 (0.00%) | 0 (0.00%) | NA |
| Penile–anal intercourse | 0 (0.00%) | 0 (0.00%) | 0 (0.00%) | NA |

BV: bacterial vaginosis, CCVR: combined contraceptive vaginal ring, COC: combined oral contraceptives, CST: community-state type, Ct: *Chlamydia trachomatis*, Ng: *Neisseria gonorrhoeae*, Tv: *Trichomonas vaginalis*, Mg: *Mycoplasma genitalium*, sd: standard deviation, IQR: interquartile range.
Chi-squared test (Fisher's exact test when expected values < 5) for the assessment of association of frequency among groups, unpaired Mann–Whitney U test for comparison of medians, and unpaired Student's t test for comparison of means.
[a]Missing data from two adolescents (COC: $n = 1$; Net-En: $n = 0$; CCVR: $n = 1$).
[b]Missing data from 26 adolescents (COC: $n = 5$; Net-En: $n = 10$; CCVR: $n = 11$).
[c]Missing data from nine adolescents (COC: $n = 2$; Net-En: $n = 5$; CCVR: $n = 2$).

**Relationship of HC with bacterial taxa and cervicovaginal cytokines.** Of the 15 cytokines measured in cervicovaginal secretions, 13 were reliably detectable, including inflammatory cytokines previously associated with HIV risk (interleukin (IL)-1β, IL-6, IFN-γ, and TNF-α), soluble CD40 ligand (sCD40L), and Th17 cytokines (since Th17 cells have been identified as key target cells for viral replication during vaginal transmission in nonhuman primates)[29,30]. As shown previously[20,21], standardized read counts of taxa associated with increased HIV susceptibility (such as *Parvimonas micra*, *Prevotella* spp., *Gemella asaccharolytica*, *Mycoplasma hominis*, and *Sneathia*) correlated positively with the concentrations of several inflammatory cytokines (Supplementary Fig. 3a). In contrast, the abundances of *L. crispatus*

and *L. iners* were negatively correlated with the concentrations of several cytokines. The canonical Th17 cytokines IL-17A and IL-17F correlated positively with abundance of bacteria not previously thought of as inflammatory in the female genital tract (FGT), including *Streptococcus anginosus*, *Staphylococcus*, and *Corynebacterium striatum* (Supplementary Fig. 3a).

In a longitudinal, paired ITT analysis from baseline to crossover, there were no significant changes in the levels of measured cytokines in the COC or Net-En arms after adjusting for multiple comparisons (Fig. 3a, b). However, the levels of IL-1β, IL-6, IL-21, IL-33, TNF-α, and IFN-γ significantly increased between baseline and crossover in adolescents randomized to CCVR (Fig. 3c and Supplementary Table 4a). Changes in IL-1β,

**Table 3 DESeq2 analysis of taxa differentially abundant between (A) randomized study arms at crossover and (B) between baseline and crossover (intention-to-treat analysis).**

| (A) | Log2FC[a] | P adj[b] | Family | Genus | Species |
|---|---|---|---|---|---|
| COC arm versus Net-En arm | −10.557 | 0.0196 | Porphyromonadaceae | Tannerella | NA |
| | −10.587 | 0.0196 | Spirochaetaceae | Treponema | vincentii |
| | −2.930 | 0.0269 | Prevotellaceae | Prevotella | amnii |
| | −22.839 | 5.92e−12 | Weeksellaceae | NA | NA |
| | −25.877 | 2.90e−15 | Prevotellaceae | Prevotella | denticola |
| | −26.790 | 3.22e−16 | Lachnospiraceae | Lachnoanaerobaculum | saburreum |
| | −26.891 | 3.22e−16 | Prevotellaceae | Prevotella | baroniae |
| | −3.933 | 0.0463 | Clostridiaceae | Clostridium | BVAB3_M_indolicus |
| | −7.012 | 0.0450 | Prevotellaceae | Prevotella | shahii |
| | −8.457 | 0.0300 | Prevotellaceae | Prevotella | oulorum |
| | −8.834 | 0.0087 | Atopobiaceae | Olsenella | NA |
| | 2.067 | 0.0321 | Lactobacillaceae | Lactobacillus | iners |
| | 3.374 | 0.0450 | Veillonellaceae | Veillonella | montpellieriensis |
| | 4.291 | 0.0071 | Lactobacillaceae | Lactobacillus | jensenii |
| | 5.046 | 0.0002 | Lactobacillaceae | Lactobacillus | johnsonii_gasseri_taiwanensis |
| COC arm versus CCVR arm | −3.086 | 0.0468 | Prevotellaceae | Prevotella | amnii |
| | −3.318 | 0.0331 | Enterobacteriaceae | NA | NA |
| | 3.875 | 0.0216 | Lactobacillaceae | Lactobacillus | johnsonii_gasseri_taiwanensis |
| | 4.242 | 0.0216 | Prevotellaceae | Prevotella | melaninogenica |
| Net-En arm versus CCVR arm | −2.485 | 0.0159 | Prevotellaceae | Prevotella | disiens |
| | −2.698 | 0.0114 | Gemellaceae | Gemella | asaccharolytica |
| | −3.288 | 0.0008 | Mycoplasmataceae | Mycoplasma | hominis |
| | −4.408 | 0.0008 | Lactobacillaceae | Lactobacillus | jensenii |
| | −4.614 | 0.0139 | Bifidobacteriaceae | Gardnerella | NA |
| | 10.862 | 0.0078 | Flavobacteriaceae | Bergeyella | NA |
| | 11.078 | 0.0064 | Lachnospiraceae | Lachnoanaerobaculum | saburreum |
| | 17.217 | 6.16e−07 | Prevotellaceae | Prevotella | baroniae |
| | 22.873 | 2.55e−12 | Prevotellaceae | Prevotella | nigrescens |
| | 24.071 | 1.53e−13 | Porphyromonadaceae | Tannerella | NA |
| | 24.324 | 9.47e−14 | Spirochaetaceae | Treponema | vincentii |
| | 24.588 | 4.90e−14 | Spirochaetaceae | Treponema | socranskii |
| | 25.338 | 2.23e−15 | Weeksellaceae | NA | NA |
| | 25.999 | 9.99e−18 | Prevotellaceae | Prevotella | oulorum |
| | 26.530 | 4.27e−16 | Actinomycetaceae | Actinomyces | NA |
| | 4.442 | 0.0334 | Spirochaetaceae | Treponema | NA |
| | 5.437 | 0.0060 | Veillonellaceae | Selenomonas | infelix |
| **(B)**[c,d] | | | | | |
| COC arm | 1.941 | 0.0142 | Lactobacillaceae | Lactobacillus | iners |
| | −11.005 | 0.0275 | Clostridiaceae | Clostridium | moniliforme |
| | −7.544 | 0.4809 | Clostridiaceae | Clostridium | perfringens |
| Net-En arm | 1.446 | 0.3449 | Prevotellaceae | Prevotella | disiens |
| | 2.002 | 0.1755 | Mycoplasmataceae | Mycoplasma | hominis |
| | 3.186 | 0.3449 | Campylobacteraceae | Campylobacter | NA |

FC: fold change, CCVR: combined contraceptive vaginal ring (A: $n = 32$, B: $n = 32$ matched participants), COC: combined oral contraceptives (A: $n = 35$, B: $n = 34$ matched participants), Net-En (A: $n = 28$, B: $n = 32$ matched participants).
[a]Positive fold changes represent taxa more abundant in the first hormonal contraceptive arm.
[b]Taxa with an adjusted $P < 0.05$ after adjusting for condom use and antibiotic use included.
[c]Positive fold changes represent taxa more abundant at crossover.
[d]Taxa with an adjusted $P < 0.5$ included.
Source data are provided as a Source Data file.

IL-6, IL-21, and TNF-α remained significant after adjusting for multiple comparisons using the Benjamini–Hochberg (BH) method. In an ITT cross-sectional analysis, seven cytokines (IL-1β, IL-6, IL-17A, IL-21, IL-25, and TNF-α) were significantly elevated in adolescents using CCVR versus adolescents using COCs. TNF-α remained significant after adjusting for multiple comparisons (adj $P < 0.05$, Supplementary Fig. 3b and Supplementary Table 4B). In a linear mixed-effect model with Tukey's post hoc test adjusted for condom use, baseline alpha diversity and antibiotic use, as well as multiple comparisons, there were no significant differences in cytokine concentrations at crossover between COC and CCVR or Net-En, but multiple cytokines were significantly higher in women assigned to CCVR versus Net-En (IL-21, -25, -31, -33, IFN-γ, and sCD40L).

**Integration of microbiota and cytokine data.** Since both Net-En and CCVR users had higher relative abundance of inflammation-associated taxa than COC users, but inflammation was the highest in CCVR, we performed multivariate logistic regression on the ITT cohort. To do this, unsupervised clustering of cytokine concentrations was used to binarize the overall genital inflammation status into inflammation-high and inflammation-low clusters (Supplementary Fig. 4a, c). The inflammation-high group had significantly higher alpha diversity compared to the inflammation-low group (median SI (IQR): 1.84 (1.00–2.22) vs. 0.83 (0.39–1.55), $P < 0.001$) and was more likely to be CST-IV (66% vs. 19%, $P < 0.001$) (Supplementary Fig. 4b). We used HC as the response variable and inflammation (high vs. low) and CST (I vs. III and IV) as our predictor variables (Supplementary Table 5). When

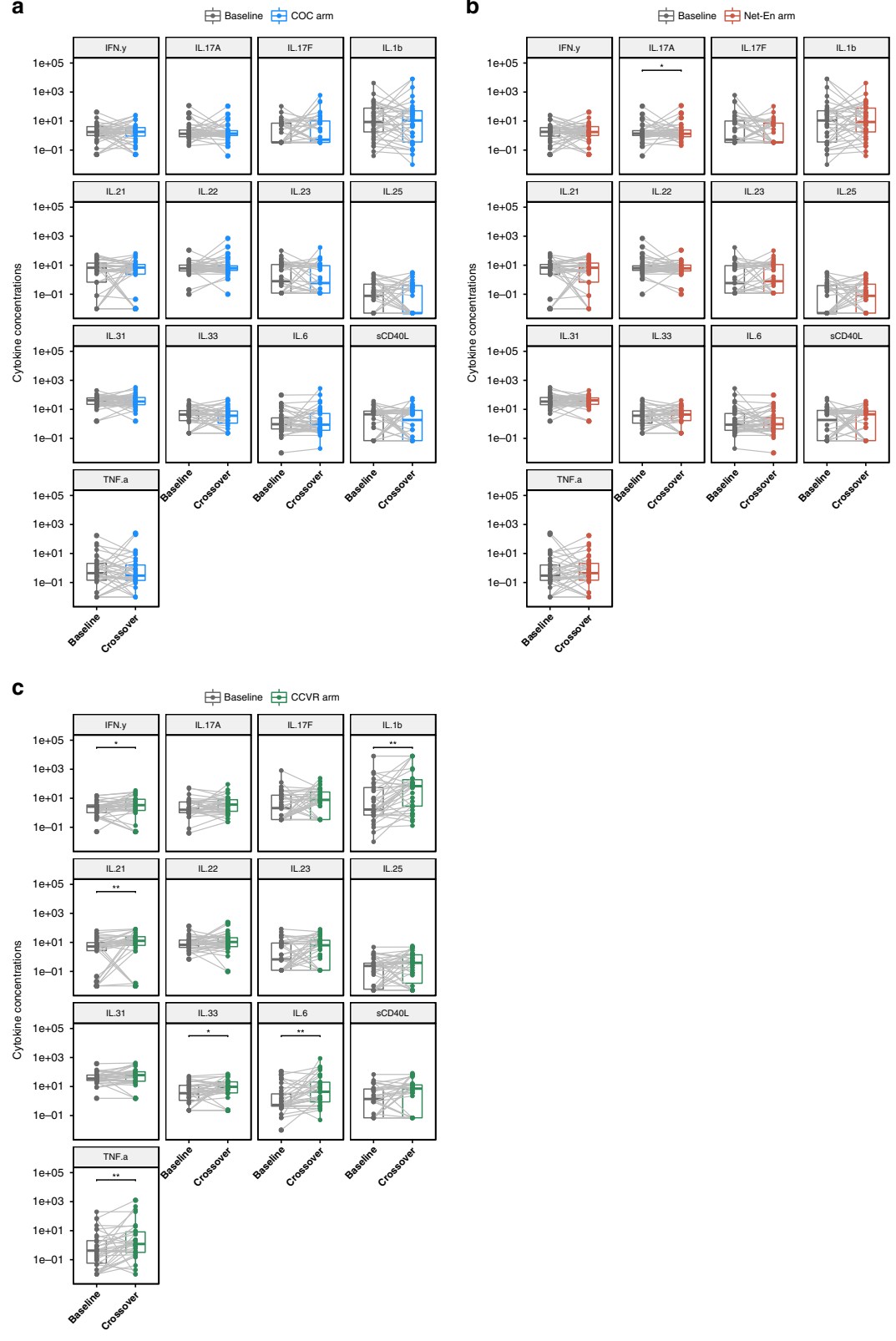

comparing adolescents in the COC arm versus those in the Net-En arm at crossover (ITT), CST-III was significantly less prevalent than CST-I in adolescents in the Net-En arm after adjustment (OR: 0.1, 95% CI: 0.02–0.5, $P = 0.009$), while CST-IV (0.5: 95% CI: 0.1–2.1, $P = 0.343$) and high inflammation (OR: 1.6, 95% CI: 0.4–6.5, $P = 0.488$) were not. When comparing adolescents in the COC arm versus the CCVR arm, there were no associations with

CST, although CCVR was associated with a nonsignficant increase in inflammation (OR: 3.1, 95% CI: 1.0–10.9, $P = 0.072$). When comparing adolescents in the CCVR and Net-En arms at crossover, no significant associations were found with inflammation group or CST (Supplementary Table 5).

We next explored the combined effect of HC on bacterial taxa and cervicovaginal cytokines implicated in HIV risk using a

**Fig. 3 Cytokine concentrations from baseline to crossover according to hormonal contraceptive arm in an intent-to-treat analysis.** Boxplot showing the change in cytokine concentrations from baseline to crossover for matched participants within each of the three contraceptive arms: **a** adolescents randomized to COC ($n = 36$), **b** adolescents randomized to Net-En ($n = 35$), and **c** adolescents randomized to CCVR ($n = 34$). P values were generated using two-sided paired Wilcoxon signed-rank tests adjusted for multiple comparisons using the Benjamini–Hochberg (BH) method. Y axis log10-transformed. Significance codes: *$P < 0.05$, **$P < 0.01$, ***$P < 0.001$. Bounds of boxes show interquartile range (IQR) with the lower and upper hinges corresponding to the 25th and 75th percentiles, respectively, lines in the middle of the box indicate median, and top and bottom whiskers demonstrate the largest and lowest value within 1.5 × IQR from the hinge. Points beyond that are plotted individually. CCVR: combined contraceptive vaginal ring, COC: combined oral contraceptives. Source data are provided as a Source Data file.

sparse Partial Least Squares Discriminant Analysis (sPLSDA). The bacterial taxa included in the analysis were *Parvimonas micra*, *Prevotella* spp. (including *P. amnii*, *P. bivia*, and *P. melaninogenica*), *G. asaccharolytica*, *Sneathia*, *V. montpellieriensis*, *M. hominis*, *Megasphaera*, *L. iners*, and *L. crispatus* due to their prior association with HIV risk or protection[9,10]. Cytokines IL-1β, IL-6, IFN-γ, and TNF-α were included in the analysis due to their prior association with HIV acquisition[19]. CCVR inititation was associated with an increase in component 1 (Fig. 4a). *P. melaninogenica*, IFN-γ, and IL-1β were all positively loaded on this component, while *P. amnii* was negatively loaded (Fig. 4a, b). Participants in the COC arm had an increase on component 2 between baseline and crossover, and *L. iners* loaded positively on component 2. On the other hand, Net-En initiation was associated with a positive shift in both components 1 and 2, associated with positive loading of *P. bivia* and *M. hominis* on component 1 and negative loading of *Sneathia* (i.e., Net-En use was associated with an increase in these taxa). However, the shifts between baseline and crossover for Net-En were small, likely because 50% were on Net-En at baseline (Table 1). Cytokine changes were not associated with either COC or Net-En arms in the sPLSDA (Fig. 4b).

## Discussion

Young African women are at high risk for both HIV infections and unintended pregnancies, and are in need of safe and effective HC. Here, we compare the effects of three HCs on vaginal bacterial communities and inflammation in a randomized trial. Due to the randomized design, we were poised to overcome many of the challenges that have plagued prior observational studies, such as confounding due to reductions in condom use by women using more effective contraceptives. Furthermore, this study focused on adolescent females, who are at the highest risk for both unintended pregnancies and HIV acquisition. We hypothesized that the progestin-only HCs would cause depletion of *Lactobacillus* in the vaginal microbiota, but that the estrogen component in both CCVR and COC would prevent this from occuring. Further, we hypothesized that *Lactobacillus*-depleted vaginal microbiota would be associated with increased inflammatory cytokines in cervicovaginal fluid. We found that adolescents assigned to COC not only had significantly less diverse vaginal microbiota compared to those assigned to Net-En, but also those assigned to CCVR. These differences were evident even when adjusting for antibiotic use as well as baseline CST. We found no evidence suggesting that the effect of HC use on vaginal alpha diveristy differed by baseline CST. Although this study was randomized, it was not blinded, and participants may have altered their sexual risk behavior due to perceived risk for pregnancy with use of different and potentially unfamiliar HC methods. Furthermore, use of HC may alter a woman's menstrual bleeding patterns contributing to the changes oberved in the vaginal microbiota with HC use. However, these potential mediators could only explain a very small and nonsignificant proportion of the effects of study arm on the microbiota. In a longitudinal, paired analysis, alpha diversity was lower when adolescents were on COC

compared to both CCVR and Net-En. Community composition differed between assigned study arms at crossover, with COC users more likely belonging to the *L. iners*-dominated CST-III type, and Net-En and CCVR users more likely to belong to the more diverse CST-IV type. Net-En users displayed a tendency to shift from an *L. iners*-dominated community toward one of the other CSTs. Taken together, these data suggest that COC may have a beneficial effect on vaginal microbiota composition. Although women randomized to COC in our study had higher relative abundance of *L. iners*, a non-$H_2O_2$-producing lactobacillus whose role in vaginal health is unclear[31,32], *L. iners* abundance was associated with lower vaginal inflammatory cytokine concentrations, and therefore in this African context, may be favorable.

While certain vaginal bacterial species have been associated with an optimal mucosal environment and protection against sexually acquired pathogens, others have been associated with increased risk of STIs[9,10]. We found that species associated with risk for HIV were more abundant in adolescents randomized to Net-En (such as *Prevotella*, *Sneathia*, and *Parvimonas*) or CCVR (such as *Prevotella*, *Mycoplasma*, and *Parvimonas*) compared to COC. In agreement with our data, a study in Rwandan female sex workers, COC users had lower semiquantitative abundance of *Prevotella*, *Sneathia/Leptotrichia amnionii*, and *Mycoplasma* species[33]. These results have implications for adolescents in SSA as COCs are rarely used, injectable contraceptives are preferred[34,35], and STI prevalence is high[36,37].

Most epidemiological data investigating the association between injectable HC and HIV susceptibility have focused on intramuscular DMPA due to the limited use and availability of Net-En[6,7]. The few studies evaluating Net-En suggest that it may have lower HIV risk implications than DMPA, possibly reflecting different synthetic progestins[38]. Different progestins could have varied effects on the FGT ecosystem, due to variations in HC dose, steroid receptor binding, peak serum progestin concentrations, and the level of estrogen suppression, among other factors[39–41]. They may therefore also have different effects on HIV susceptibility. The recently completed Evidence for Contraceptive Options and HIV Outcomes (ECHO) Study[42], which compared DMPA, copper intrauterine device (IUD), and levonorgesterol (LNG) implant, found no increased risk of HIV with DMPA-IM use compared to Copper-IUD or LNG implant. However, ECHO did not include any of the HC we studied here. Furthermore, ECHO did not include a contraceptive method with an estrogen component and therefore did not address the potential effects of topical or systemic estrogen. The high HIV incidence in contracepting women in the ECHO trial highlighted the urgent need to expand contraceptive method options and integrate HIV and sexual and reproductive health products and services, and therefore evaluation of a broad range of contraceptive options is an urgent public health priority.

There are limited data on the effects of CCVRs on vaginal microbiota composition, with most studies using Nugent scoring and culture-based methods[43–45]. In a recent open-label study, the impact of CCVR use on vaginal microbiota composition was evaluated pre- and 12 weeks post CCVR initiation by qPCR of

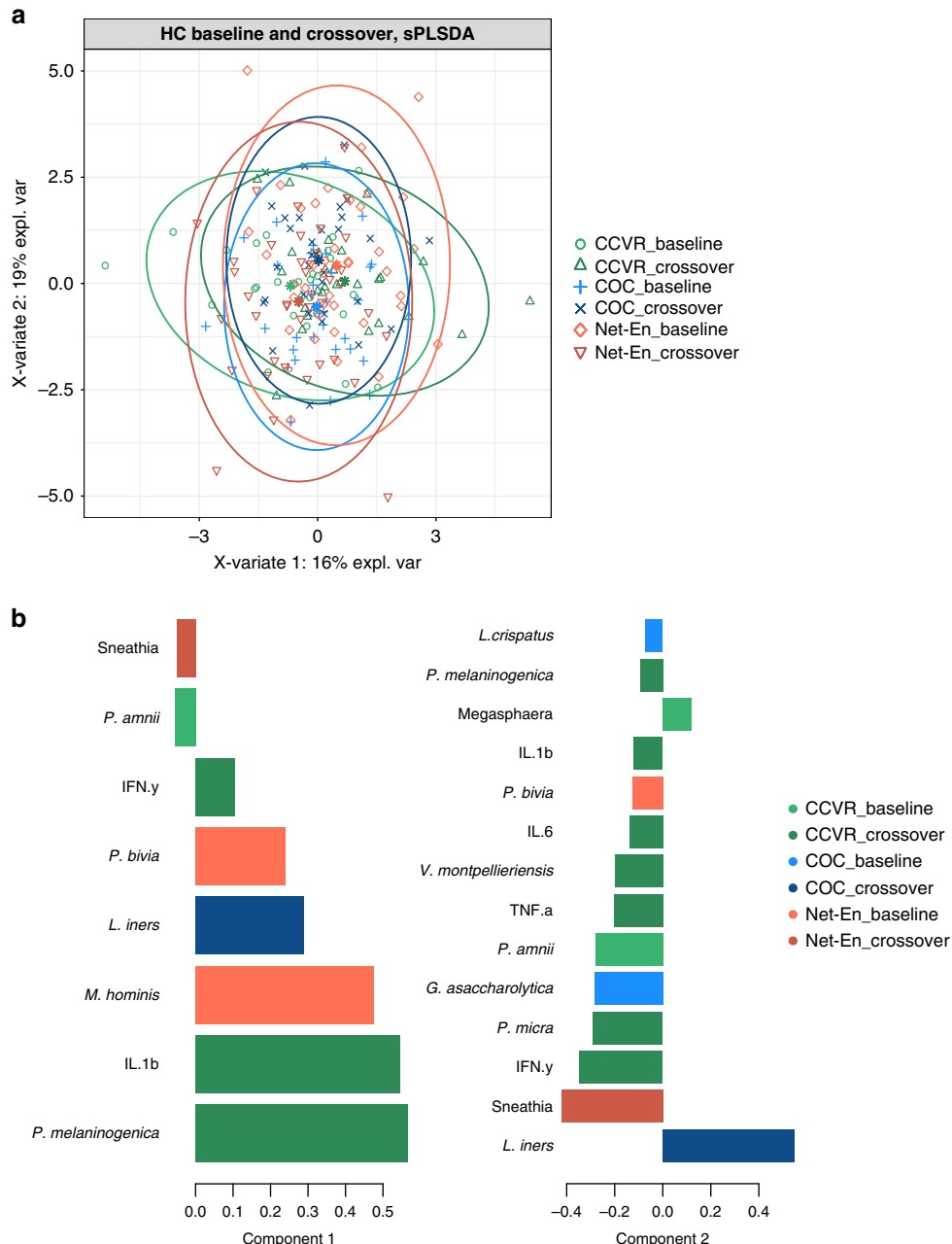

**Fig. 4 Effect of hormonal contraception on bacterial taxa and cervicovaginal cytokines implicated in HIV risk. a** A multilevel sparse partial least-squares discriminant analysis (sPLSDA) of cytokines and bacterial taxa associated with HIV grouped before (baseline) and after (crossover) use of COC, Net-En, or CCVR, respectively. **b** Barplot showing the loadings of components 1 and 2 from the sPLSDA analysis. The absolute value is an indication of the importance of the bacteria or cytokine, while the sign indicates positive/negative correlations between the variables. CCVR: combined contraceptive vaginal ring, COC: combined oral contraceptives. Source data are provided as a Source Data file.

key vaginal bacteria (i.e., *Lactobacillus* species, *Gardnerella vaginalis*, and *Atopobium vaginae*) in 120 Rwandan women[44]. The presence and concentrations of *Lactobacillus* species in vaginal fluid increased significantly, whereas the concentration of *G. vaginalis* and presence of *A. vaginae* decreased after initiation of CCVR, suggesting that the CCVR favored healthy vaginal microbe colonization over BV-associated anaerobes[44]. In our present study, measuring relative abundance rather than absolute quantities of bacteria, we observed the opposite. It is likely that both the relative abundance and the absolute concentrations of key bacteria are important factors for optimal vaginal health. Of more concern was our finding that several cervicovaginal cytokines (including those that could put women at risk for later HIV

seroconversion[19]) were significantly elevated in adolescents randomized to CCVR compared to either COC or Net-En. In a sPLSDA analysis of cytokines and bacterial taxa associated with HIV risk or protection, CCVR use was associated with increases in both cytokines and bacteria associated with risk, while Net-En and COC use was associated with HIV risk-modifying bacteria only (*P. bivia, M. hominis, and Sneathia* for Net-En and *L. iners* for COC), possibly explaining the impact of CCVR use on genital inflammation.

Since the vaginal microbiota of adolescents initiating both Net-En and CCVR included higher relative abundance of bacterial species associated with genital inflammation, as well as diverse communities, the lack of inflammation in women intiating Net-

En was surprising. Although Net-En users had higher relative abundance of inflammatory taxa, they also had a high relative abundance of *L. crispatus*, which may dampen any inflammation. Elevated cervicovaginal cytokines in adolescents using CCVRs could reflect an inflammatory cytokine response to the presence of a foreign body in the FGT or to biofilm formation on the ring[45]. Considering both changes in genital cytokines and microbial diversity associated with HC arm, a multivariate logistic regression analysis suggested that the difference in the diversity of the vaginal microbiota was more profound between study arms than cytokine inflammation categories. The CCVR group also had a higher number of *Candida* infections compared to the other groups. In Rwandan women, the percentage of women with vaginal yeast increased from 5 to 22% after CCVR initiation[46], and in vitro data corroborate this[47], further supporting this possibility. Many different vaginal rings are currently in clinical trials, and our findings may be specific to NuvaRing®. Furthermore, our study did not test the effects of continuous-use CCVR, which may be associated with less bleeding and therefore increased quantities of *Lactobacillus*[48].

One limitation of this study was the use of the V4 region for 16S sequencing. The V4 regions did not allow for distinguishing some species-level OTUs, including lactobacilli species such as *L. johnsonii*, *L. gasseri*, and *L. taiwanensis*. Primers spanning the V3/V4 region would identify more species-level resolution in the vaginal microbiota than V4 alone. Our study was also limited by the number of participants enrolled and the loss-to-follow-up rates; however, the longitudinal study design provided us with the opportunity to perform both within-subject assessments comparing the time periods before and after randomization to one of the three HCs in addition to the cross-sectional comparisons of the difference between the study interventions. Due to the study design, participants randomized to COC could have received one of two COC that differed in their components. However, if anything, this would have introduced bias toward to null. Adherence to the randomization arm tended to be poorer for those using the ring, as has been reported in other trials[49]. However, this study differs from many of these prior ring studies as the adolescents in this study were seeking effective contraception and knew that they had the risk of pregnancy with nonadherence. Additionally, since the aim of the parent study was to assess feasibility and acceptability of methods in adolescents, reporting of problems with adherence and method changes in response were encouraged, albeit reliant on self-report. The study did not include an HC-unexposed group, and the screening visit in most cases did not represent a true, contraceptive-naive baseline control for comparison with no washout periods. Although highly variable, NET can been detected in blood for 6 months after discontinuation of Net-En injection, although at lower levels than needed for contraceptive efficacy[50]. However, randomization ensured that there was equal distribution between arms of previously used methods of contraception at crossover. Nevertheless, it is not known what impact these low levels of NET have on microbiota and inflammation in the presence of other hormones. Future randomized trials with a larger number of contraceptive-naive adolescents would be helpful to further study the biological impact of HC on FGT microbiota composition. Regardless, insights gained from this study can potentially contribute to the development of expanded, safe contraceptive options for young women in HIV-prevalent settings.

## Methods

**Study cohort.** Adolescents were recruited through a parent study, UChoose, an open-label, randomized crossover study designed to evaluate the feasibility of different HC options among adolescents[25]. The parent study was approved by the Division of AIDS and the University of Cape Town (UCT) Health Science Research Ethics

Committee and was conducted in full compliance with South African Good Clinical Practice (SA-GCP), ICH76 GCP guidelines, and ICMJE guidelines, and registered in the public registry database of ClinicalTrials.gov (NCT02404038). Approval for this substudy was obtained from the Human Research Ethics Committee at the University of Cape Town (HREC 801/2014). Participants were screened, those 18 years or older provided informed consent for the substudy, while assent from the participant and informed consent from a parent or legal guardian were obtained for participants younger than 18 years old. Eligibility criteria for enrollment for the parent study included either HC naive or willingness to change the method, no symptomatic STIs within the prior 40 days, no known sensitivity to any of the study products, and no intentions of becoming pregnant throughout the study period. Furthermore, participants were asked to abstain from inserting any nonstudy products or objects into the vagina throughout the duration of study. The parent study enrolled 130 HIV-negative, nonpregnant adolescent girls aged 15–19 years between September 22, 2015 and June 30, 2017, all of whom consented to participate in this substudy. Participants returned for enrolment within 40 days of the screening visit. At enrolment, participants were randomly assigned in a 1:1:1 ratio to one of three study arms for a 16-week period: Arm 1: injectable hormonal contraception (Net-En, containing 200 mg of the progestogen norethisterone enantate) once every 8 weeks, arm 2: combined contraceptive intravaginal ring (CCVR, NuvaRing®; MSD Pty Ltd, containing etonogestrel/ethinyl estradiol, 0.120 mg/0.015 mg per day) to be inserted once every 28 days (and removed after 21 days of each 28- day insertion), or arm 3: combined oral contraceptive pills (COC) (either Triphasil® or Nordette®, both containing ethinyl estradiol/levonorgestrel (Triphasil® (triphasic regimen): six tablets containing 30 µg/50 µg, five tablets containing 40 µg/75 µg, and ten tablets containing 30 µg/125 µg; Nordette® (monophasic regimen): all tablets contain 30 µg/150 µg daily for 21 days each month with a placebo tablet for 7 days (days 22–28)). After 16 weeks, participants returned at which time they switched over to another HC option for the final 16 weeks of the study. Participants receiving either Net-En or COC for the first 16 weeks of the study (arms 1 and 3) switched over to the CCVR, while those receiving the CCVR as the first method (arm 2) were given the choice between either Net-En or the daily COCs as their second method. After a further 16 weeks, participants returned for a final visit at the clinic and exited the study. Randomization was performed using random number sequence in Stata and provided to the pharmacist in sealed envelopes. Safety was measured by monitoring laboratory and clinical adverse events using the DAIDS table for grading the severity of adult and pediatric adverse events, version 1.0, December 2004 (clarification August 2009).

**Power calculations.** The sample size of the substudy was limited to that of the parent study, where power calculations were based on the study's primary outcome, the relative acceptability of CCVR versus other modalities based on the total score for the ORTHO BC SAT questionnaire[51] at 4 months after randomization. Power for the substudy was calculated for the aim of evaluating CD4 + cervical T-cell populations at 4 months post contraceptive method initiation[52] as prior data on the microbial endpoints were not available. Based on estimates from previous research[53,54], $n = 50$ per group provides 80% power to detect absolute differences of 10% in the mean frequencies of CD4 + HLADR + CD38 + cervical T cells between any two contraceptive groups by one-way ANOVA at alpha = 0.05.

**Sample collection.** At all study visits, a rapid HIV and a pregnancy test were performed and if positive, the participant was counseled and referred for management, and no further mucosal samples were collected. A detailed interviewer-assisted questionnaire assessing medical history, sexual behavior, the last menstrual cycle, adherence to study the product, intravaginal practices, adverse experiences, and antibiotic use was completed. The following genital tract samples were collected at the screening, crossover, and exit-study visits: cervical secretions collected using a Softcup® menstrual cup inserted for 30 min; two vulvo-vaginal swabs for STI testing, Nugent scoring, *Candida* screening, and pH measurement, and a lateral wall swab for 16S rRNA gene sequencing. Upon arrival at the laboratory, vaginal swabs and menstrual cup secretions were stored at −80 °C until testing. No samples were collected during menstruation; instead the visit was rescheduled.

**STI and BV testing and treatment.** Molecular testing for the following STIs: *Chlamydia trachomatis*, *Neisseria gonorrhoeae*, *Trichomonas vaginalis*, and *Mycoplasma genitalium* by multiplex PCR was performed at each visit as described[55]. If any of these laboratory-based tests were positive, appropriate targeted therapy was prescribed and recorded. Blood was obtained for HIV rapid test and herpes simplex virus 2 (HSV-2) serology. A vulvo-vaginal swab was collected for BV testing (Gram staining and Nugent scoring; BV negative (Nugent 0–3), intermediate (Nugent 4–6), or positive (Nugent 7–10)), and microscopy for *Candida* hyphae and spores. Treatment for BV was only offered to participants presenting with clinical symptoms according to South African syndromic management guidelines. Vaginal pH was measured using color-fixed indicator strips (Macherey-Nagel, Düren, Germany).

**Sequencing of the V4 region of the 16S rRNA gene.** Vaginal lateral wall swabs were collected for microbiome analysis using 16S rRNA gene sequencing. Swabs from 130 participants from 329 sample visits were thawed and treated with an enzyme cocktail consisting of mutanolysin (25kU/ml, Sigma Aldrich), lysozyme (450 kU/ml, Sigma Aldrich), and lysostaphin (4 kU, Sigma Aldrich) for 1 h at

37 °C. Microbial DNA was extracted using the Quick-DNA™ Fungal/Bacterial Miniprep kit (Zymo Research) following the manufacturer's protocol. Mechanical disruption was performed in a Qiagen TissueLyser LT for 5 min at 50 oz. The V4 hypervariable region of the bacterial 16 S rRNA gene was amplified by PCR using modified universal primers[56]: 515 F (5′-TCG TCG GCA GCG TCA GAT GTG TAT AAG AGA CAG NNN NNG TGC CAG CMG CCG CGG TAA-3′) and 806 R (5′-GTC TCG TGG GCT CGG AGA TGT GTA TAA GAG ACA GNN NNN GGA CTA CHV GGG TWT CTA AT-3′) (Supplementary Table 6). For each PCR reaction, 1 μl of 515 F primer (5 μM), 1 μl of 806 R primer (5 μM), 12.5 μl, KAPA Hotstart ReadyMix, template DNA (10–20 ng/μl), and molecular-grade water was used for a final volume of 25 μl. PCR was performed in a thermal cycler (Geneamp PCR system 9700, Applied Biosystems) using the following program: 95 °C for 3 min, 35 cycles of 95 °C for 30 s, 55 °C for 30 s, 72 °C for 30 s, and 72 °C for 5 min. Samples were purified using Agencourt AMPure XP beads (Beckman Coulter, Brea, CA, USA) and quantified using the Qubit dsDNA HS Assay (Life Technologies) according to the manufacturer's protocols. Illumina sequencing adapters and dual-index barcodes were added to the purified amplicon products using limited-cycle PCR and the Nextera XT Index Kit (Illumina) (Supplementary Table 6). For each PCR reaction, 5 μl of forward primer (1 μM), 5 μl of reverse primer (1 μM), 25 μl of KAPA Hotstart ReadyMix, 5 μl of template DNA (amplicon products from the first PCR), and 10 μl of molecular-grade water was used. Limited-cycle PCR was performed using the following program: 95 °C for 3 min, eight cycles of 95 °C for 30 s, 55 °C for 30 s, 72 °C for 30 s, and 72 °C for 5 min. Amplicons from 96 samples and positive (bacterial mock community) and negative (DNA extraction and PCR water) controls were pooled in equimolar amounts, and the resultant libraries purified by gel extraction (Qiagen) and quantified using the Qubit dsDNA HS Assay Kit (Life Technologies). The libraries were sequenced on the Illumina MiSeq platform (300-bp paired end) with v3 chemistry.

**Bioinformatics analysis of the 16S rRNA gene-sequencing data.** The sequencing reads were analyzed using the following in-house analysis pipeline (https://github.com/uct-cbio/16S-rDNA-pipeline). Following demultiplexing, raw reads were preprocessed as follows: forward and reverse reads were merged using usearch7[57], allowing a maximum of three mismatches and the resultant merged reads were quality-filtered using usearch7. Primer sequences were removed using a custom python script and reads truncated at 250 bp. Sequences were then dereplicated and clustered de novo into operational taxonomic units (OTUs) at 97% similarity using usearch7. Chimeric sequences were detected (against the Gold database) using UCHIME[58] and removed. Individual sequences were assigned to the specific identifiers using a 97% similarity threshold. Taxonomic assignment was performed in QIIME 1.8.0[59] using the RDP classifier (using the default confidence level of 0.5) against the GreenGenes 13.8 reference taxonomy. To increase species-level resolution, we used the usearch_global command implemented in VSEARCH[60] to search the de novo-picked OTUs' representative sequences against our own Custom Vaginal 16 S Reference Database previously described[21,61]. All hits with ≥97% identity were accepted. The remaining OTUs (<97% identity) were manually curated using BLAST on NCBI's nucleotide database (excluding uncultured organisms). OTUs that mapped to more than one species (with the same identity score) were annotated as follows: if an OTU mapped to two or more species, the OTU would be named Genus speciesA_speciesB or Genus speciesA_speciesB_speciesC, respectively; if an OTU mapped to more than three species but one species was clearly associated with vaginal microbiota (based on prior knowledge), the OTU was named Genus species_cluster, where "species" was selected based on the majority of hits[21]. One OTU that mapped to both *L_crispatus* and *L. acidophilus* will be referred to as *L. crispatus* throughout this paper due to its known presence in the vagina, despite us not being able to distinguish between these two species using the V4 region of the 16 S rRNA gene. The results were imported as a phyloseq object in R[62]. Samples with ≥5000 reads were selected for downstream analyses. The OTU table was normalized (i.e., transformed to relative abundance * median sample read depth), and filtered so that each OTU had to have at least ten counts in at least 20% of samples or have a relative abundance of at least 0.001% unless otherwise specified.

**Cytokine measurements.** The concentrations of IL-1β, IL-4, IL-6, IL-10, IL-17A, IL-17F, IL-21, IL-22, IL-23, IL-25, IL-31, IL-33, IFN-γ, sCD40L, and TNF-α in Softcup® cervical secretions were measured by Luminex Milliplex assay using the Bio-Plex Pro Human Th17 Cytokine Panel. The assay plates were read using a Bio-Plex Suspension Array Reader (Bio-Rad Laboratories Inc., USA) and Bio-plex manager software version 4 (Bio-Rad Laboratories, Inc.). A 5-parameter logistic regression formula was used to calculate sample concentrations from the standard curves. For samples with cytokine values below the limit of detection, half of the lowest-detectable value for the specific cytokines was assigned. For samples with values above the detection limit, the highest-detectable value was applied. Specimens from six participants were run across all plates (interplate controls), and samples from six participants were duplicated on each set of plates (intraplate controls) for quality control measures. Spearman's rank test was used to measure intra-assay and inter-assay correlation coefficients to determine assay reliability and reproducibility. For the individual cytokines, a cutoff of 55% detectable samples was set for inclusion in analyses. Unsupervised clustering of the cytokine data was performed by partitioning around medoids (PAM) clustering

available in the R package "cluster"[26] with Euclidean distances and optimal k (the k producing the maximum average silhouette width).

**Statistical analyses.** All downstream statistical analyses were performed in RStudio using the packages phyloseq[63] for beta-diversity analyses with weighted UniFrac distances, vegan[64] for alpha-diversity estimates (Shannon Index), ordinations, and redundancy analysis. Microbiota community clusters were established by Fuzzy clustering using the R package "cluster"[26] with optimal k, a membership exponent of 1.25, and weighted UniFrac as the dissimilarity measure. Differences in study population characteristics according to study arm were tested using Pearson's Chi-squared test or Fisher's exact test (when the expected value was <5) for count data and unpaired Student's $t$ test for differences in mean (parametric data) and unpaired Mann–Whitney $U$ test for differences in medians (nonparametric data) with post hoc testing. Linear regression models were used to adjust for covariates such as antibiotic use and baseline CST. Baseline CST was also included as an interaction term in order to evaluate the impact of HC on the microbiota according to entry microbiota. Mediation analysis was conducted using the R package "mediation" comparing two arms at the time[65]. Multivariate logistic regression analyses with Tukey's post hoc testing were used to examine the association of study arm with alpha diversity and inflammation category. Linear mixed-effect regression models for continous data and conditional logit models of binary responses and multinomial counts were used for repeated measure analysis, and Wilcoxon signed-rank and Friedman tests were used for paired, longitudinal analyses. The overall difference in microbial composition between groups was determined by permutational multivariate analysis of variance (PERMANOVA) using distance matrices (weighted UniFrac) with 999 permutations using the adonis2 function; homogeneity of variance between groups was assessed using the betadisper function using vegan[64]. OTUs were first merged at the lowest available taxonomic level using a custom script[21]. Spearman's rank test was applied to test for correlation between nonparametric data. The R function nominalSymmetryTest in the "rcompanion" package[66] was used to conduct omnibus symmetry exact tests for the CST transitions between baseline and crossover in each study arm. The model input included the participant ID, CST, and follow-up time in weeks for each sample, and study arm was modeled as a covariate. The study was not powered to calculate the 95% confidence interval to compare the study arms. Differential abundance analysis was performed using DESeq2[67]. OTUs with an adjusted $P$ value of <0.5 and an estimated fold change of >1.5 or <1/1.5 were included. Sparse partial least-squares discriminant analysis (sPLSDA) was used for variable selection using the "mixOmics" package in R[68]. The input variables used for sPLSDA analysis included log ratio (CLR)-transformed bacterial count data and log-transformed cytokine levels before (baseline) and after (crossover) HC use.

**Reporting summary.** Further information on research design is available in the Nature Research Reporting Summary linked to this article.

## Data availability

Raw sequence data for 16S rRNA gene amplicon sequences are available at http://www.ebi.ac.uk/ under project number PRJEB30774. R analysis scripts are available on http://github.com/frk.balle/uCHOOSE. The parent study protocol can be found at https://clinicaltrials.gov/ct2/show/record/NCT02404038. Source data are provided with this paper.

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

## Acknowledgements

This study was supported by grants from South African MRC and National Institutes of Health (R01 HD083040 for substudy to H.B.J. and J.-A.S.P., and R01AI094586 for parent study to L.G.B.). The DTHF also recognizes the support from MSD PTY LTD for the donation of NuvaRings®. MSD did not take part in study design, data collection, analysis, or paper writing. Part of this work was performed at Lawrence Berkeley National Laboratory, managed by the University of California for the U.S. Department of Energy under Contract No. DE-AC02-05CH11231. We thank the UChoose study team, particularly Pinky Ngobo, Janine Nixon, Eve Mendel, and Keshani Naidoo and all the young women who kindly participated in the study. We would also like to thank Thandi Magwai, Wanani Mubilanzila, Valerie Masete, and Madeleine Heller for their contributions in the laboratory. Computations were performed using facilities provided by the University of Cape Town's ICTS High Performance Computing team: http://hpc.uct.ac.za. C.B. was supported in part by the Poliomyelitis Research Foundation of South Africa.

## Author contributions

Conceived and designed the experiments: H.B.J., J.-A.S.P., C.B., S.E.B., L.M., U.K., and E.B. Designed and recruited the uCHOOSE cohort: L.G.B., K.G., T.P., and S.L.B. Processed samples and performed wet-lab experiments: C.B., S.J., I.K., R.E., E.H., A.H., and H.G. Performed STI and BV testing: V.M. Analyzed the data: C.B., K.L., Z.M., and H.B.J. Wrote the paper: C.B., J.-A.S.P., and H.B.J.

## Competing interests

The authors declare no competing interests.
