## [Peer Review File · Nature Communications]

Reviewers' comments:

Reviewer #2 (Remarks to the Author):

These authors followed 131 (or 130?) HIV negative adolescent girls aged 15-19 recruited from the Desmond Tutu HIV Foundation Youth Centre in Masiphumelele, Cape Town, South Africa, were placed on one of three hormonal contraceptives: 1) long-acting injectable norethisterone enanthate (Net-En); 2); combined oral contraceptives (COC); or combined contraceptive vaginal ring (CCVR). After 16 weeks girls on COCs or CCVRs were switched to Net-En, and girls on Net-En were switched to either COCs or CCVRs for an additional 16 weeks. Vaginal wall samples were collected at enrollment, at the switch visit, and after the final 16 weeks, and subjected to taxonomic analysis by 16S rRNA sequencing and cytokine profiling. 107 subjects were sampled in the switch visit, and 92 remained at the 16 week follow up visit. The main

observations were that: 1) The bulk of the samples were classified into Community States I, III or IV, consistent with the classification scheme of Ravel et al. This was a suggestion of the previous reviewers and improves the manuscript. 2) samples from women on COCs, whether in the switchover visit sample or the exit sample, had a lower complexity, more CS III-like *L. iners* dominated microbiome profiles, in contrast to samples from women using CCVR and Net-En contraception, who were generally more likely to have a more diverse community; 3) some bacterial taxa were more significantly differentially abundant among young women with different contraceptives (*L. iners* more and *Sneathia* and *BVAB3* less in COC compared to Net-En; *Anaerococcus prevotii_tetradius*, *Enterobacteriaceae* and *Fusobacterium equinum* less in COC vs CCVR; and *P. bivia* and *BVAB3_M_indolicus* more abundant in CCVR vs Net-En; at the switch visit); 4) vaginal cytokines were elevated in participants who received CCVR compared to the other women who received COCs or Net-En.

The main conclusion of the research is that COCs, relative to CCVR or Net-En contraceptive, exert a positive influence on health of the FGT by elevating prevalence of *Lactobacillus* while reducing overall diversity of the microbiome at the expense of taxa associated with inflammation. The overall project seems quite solid from most perspectives. The abstract is clearly written, as is the rest of the manuscript, the conclusions seem solid and appropriate.

Critique

The manuscript has been extensively rewritten and improved. The authors have responded adequately to the previous review. Whereas the results are not completely novel, this topic is quite important and yet still controversial and needs to be explored in more depth.

Below are listed a number of minor issues I noticed in the text.

Page 7: fix “driven primarily be the differences”

Page 12: did not understand this: “because although this study was randomized, it was not blinded, and participants may have altered behaviour due to perceived risk for pregnancy”

Page 13: fix: “although women randomized to COC in our study had higher relative abundance of L. iners, a non H₂O₂-producing (taxon?) whose role in vaginal health is unclear”

Fix: “with a optimal”

Cumbersome: “significantly more relatively abundant”

Table 3: please use small case for species names.

Fig. 11. Labeling is mixed up (no A or B), only C. C should be B.

Page 14: Unclear “expand contraceptive method (mix?) and integrate HIV and sexual and reproductive health products and services”

Page 14: which bacteria? “COC use was associated with bacteria only, thus underlying the potential impact of CCVR use on inflammation.”

Page 14: increased? “which may be associated with less bleeding and therefore increase quantities of Lactobacillus”

Reviewer #4 (Remarks to the Author):

Responses to Reviewer #1:

The authors responded to Reviewer #1 that the revised manuscript focused on pre-post changes rather than just cross-sectional comparisons. Detailed below are a number of issues that remain unclear about the new longitudinal analyses:

1. Significance codes “*: $p < 0.05$, **: $p < 0.01$, ***: $p < 0.001$ ” are noted at the bottom of Figure 2 but it appears that only Figure 2H was marked as $p < 0.05^*$.
2. Line 136-138 states “a decrease in alpha diversity was observed in the COC and Net-En arms from baseline to crossover, while an increase in the CCVR arm was observed (Figure 2E-G)”. However it doesn’t appear that statistics are presented to confirm this finding in Figure 2. There are no asterisks per the above key?
3. Line 144-147: the linear regression model adjusting only for baseline alpha diversity demonstrated that alpha diversity in the COC arm was significantly lower than the CCVR arm (and not Net-En). It is unclear why the authors presented a model that only adjusted for baseline alpha diversity. There are a number of important individual-level confounders that are not addressed, such as antibiotic use (symptomatic bacterial vaginosis treatment or STD treatment are detailed in the methods), menstrual cycle information, sexual behaviors such as condoms, vaginal lubricant, as well as Nonoxynol-9. All of these factors may affect both the vaginal microbiota and host immune response and are therefore potential confounders to the analysis. The authors assert that there were no reported differences in sexual risk behavior or number of days since last menstrual period between the study arms, but what they assert is aggregate-level information which does not address the individual level information. If they have the individual-level variables, why not assess them in full modeling? In line 174, the authors included condom use in the models for the per protocol (PP) analysis? This appears as haphazard modeling.
4. In line 149, the authors conduct a multivariate analysis but only include baseline alpha diversity in their model.
5. Table 3 does not present a multivariate model with confounders to assess changes in taxa. They simply evaluate differentially abundant taxa between and among women.

Overall, while the manuscript is improved to present paired pre-post statistics and figures, the longitudinal modeling could be more rigorous. The authors have an opportunity to set the standard for not only strong study design (randomized and cross-over study) but also they can present robust statistical modeling and assessments. The authors should expand the longitudinal modeling for all outcomes (microbiota and cytokine data).

Reviewer #5 (Remarks to the Author):

The manuscript has addressed almost all of the reviewers concerns.

The authors still have not addressed spelling out of abbreviations on first use (Net-EN for example on line 77, COC on line 69, CCVR line 78, CSTs line 104, SPLSD line 260 etc.

Figure 1 needs further formatting so that the boxes do not cover the text. Similarly Supplemental figure 3 also has formatting issues (but this may be the electronic version)

Reviewers' Comments:

Reviewer #2 (Remarks to the Author):

These authors followed 131 (or 130?) HIV negative adolescent girls aged 15-19 recruited from the Desmond Tutu HIV Foundation Youth Centre in Masiphumelele, Cape Town, South Africa, were placed on one of three hormonal contraceptives: 1) long-acting injectable norethisterone enanthate (Net-En); 2) combined oral contraceptives (COC); or combined contraceptive vaginal ring (CCVR). After 16 weeks girls on COCs or CCVRs were switched to Net-En, and girls on Net-En were switched to either COCs or CCVRs for an additional 16 weeks. Vaginal wall samples were collected at enrollment, at the switch visit, and after the final 16 weeks, and subjected to taxonomic analysis by 16S rRNA sequencing and cytokine profiling. 107 subjects were sampled in the switch visit, and 92 remained at the 16 week follow up visit. The main observations were that: 1) The bulk of the samples were classified into Community States I, III or IV, consistent with the classification scheme of Ravel et al. This was a suggestion of the previous reviewers and improves the manuscript. 2) samples from women on COCs, whether in the switchover visit sample or the exit sample, had a lower complexity, more CS III-like *L. iners* dominated microbiome profiles, in contrast to samples from women using CCVR and Net-En contraception, who were generally more likely to have a more diverse community; 3) some bacterial taxa were more significantly differentially abundant among young women with different contraceptives (*L. iners* more and *Sneathia* and *BVAB3* less in COC compared to Net-En; *Anaerococcus prevotii*_tetradius, *Enterobacteriaceae* and *Fusobacterium equinum* less in COC vs CCVR; and *P. bivia* and *BVAB3_M_indolicus* more abundant in CCVR vs Net-En; at the switch visit); 4) vaginal cytokines were elevated in participants who received CCVR compared to the other women who received COCs or Net-En.

The main conclusion of the research is that COCs, relative to CCVR or Net-En contraceptive, exert a positive influence on health of the FGT by elevating prevalence of *Lactobacillus* while reducing overall diversity of the microbiome at the expense of taxa associated with inflammation. The overall project seems quite solid from most perspectives. The abstract is clearly written, as is the rest of the manuscript, the conclusions seem solid and appropriate.

Critique

The manuscript has been extensively rewritten and improved. The authors have responded adequately to the previous review. Whereas the results are not completely novel, this topic is quite important and yet still controversial and needs to be explored in more depth.

Below are listed a number of minor issues I noticed in the text.

Page 7: fix "driven primarily be the differences"

This has been corrected.

Page 12: did not understand this: "because although this study was randomized, it was not blinded, and participants may have altered behaviour due to perceived risk for pregnancy"

The sentence has been clarified.

Page 13: fix: “although women randomized to COC in our study had higher relative abundance of *L. iners*, a non H₂O₂-producing (taxon?) whose role in vaginal health is unclear”

This has been corrected (lactobacillus)

Fix: “with a optimal”

This has been corrected.

Cumbersome: “significantly more relatively abundant”

“Significantly” has been removed.

Table 3: please use small case for species names.

This has been corrected.

Fig. 4. Labeling is mixed up (no A or B), only C. C should be B.

This has been corrected.

Page 14: Unclear “expand contraceptive method (mix?) and integrate HIV and sexual and reproductive health products and services”

This has been clarified to read “contraceptive method options”

Page 14: which bacteria? “COC use was associated with bacteria only, thus underlying the potential impact of CCVR use on inflammation.”

*Sentence have been changed to include specific taxa: “while Net-En and COC use was associated with HIV-risk modifying bacteria only (*P. bivia*, *M. hominis* and *Sneathia* for Net-En and *L. iners* for COC), thus underlying the potential impact of CCVR use on inflammation.”*

Page 14: increased? “which may be associated with less bleeding and therefore increase quantities of Lactobacillus”

This has been corrected.

Reviewer #4 (Remarks to the Author):

Responses to Reviewer #1:

The authors responded to Reviewer #1 that the revised manuscript focused on pre-post changes rather than just cross-sectional comparisons. Detailed below are a number of issues that remain unclear about the new longitudinal analyses:

1. Significance codes “*: $p < 0.05$, **: $p < 0.01$, ***: $p < 0.001$ ” are noted at the bottom of Figure 2 but it appears that only Figure 2H was marked as $p < 0.05^*$.

*Correct. **: $p < 0.01$, ***: $p < 0.001$ has been removed.*

2. Line 136-138 states “a decrease in alpha diversity was observed in the COC and

Net-En arms from baseline to crossover, while an increase in the CCVR arm was observed (Figure 2E-G)". However it doesn't appear that statistics are presented to confirm this finding in Figure 2. There are no asterisks per the above key?

This was not statistically significant, which have now been highlighted in the text.

3. Line 144-147: the linear regression model adjusting only for baseline alpha diversity demonstrated that alpha diversity in the COC arm was significantly lower than the CCVR arm (and not Net-En). It is unclear why the authors presented a model that only adjusted for baseline alpha diversity. There are a number of important individual-level confounders that are not addressed, such as antibiotic use (symptomatic bacterial vaginosis treatment or STD treatment are detailed in the methods), menstrual cycle information, sexual behaviors such as condoms, vaginal lubricant, as well as Nonoxynol-9. All of these factors may affect both the vaginal microbiota and host immune response and are therefore potential confounders to the analysis. The authors assert that there were no reported differences in sexual risk behavior or number of days since last menstrual period between the study arms, but what they assert is aggregate-level information which does not address the individual level information. If they have the individual-level variables, why not assess them in full modeling?

The reason we only included baseline alpha diversity is that in an intent-to-treat analysis of a randomised trial, there should be balanced individual level confounders at baseline. However, for some reason, alpha diversity was not and therefore was included in the model. None-the-less, given that the trial was not blinded, and some baseline characteristics may have changed during follow-up, we have included a model that takes other individual-level confounders including antibiotic use and sexual risk behaviour (condom use) - into account in addition to baseline alpha diversity. Menstrual data was very incomplete as most adolescents reported not remembering, rendering us with few women in the models. Intravaginal practices were extremely rare in this cohort (n=7), therefore unlikely to be a contributor. Since BV and vaginal microbiota, including alpha diversity, are highly collinear, BV was not included in the model; likewise for antibiotic use and STIs (antibiotics were prescribed for laboratory diagnosed STIs).

In line 174, the authors included condom use in the models for the per protocol (PP) analysis? This appears as haphazard modeling.

Apologies if this appeared haphazard – we assure you that it was not, but rather it was because reported condom use differed according to arm at exit. However, we have included a model that takes additional possible individual-level confounders as in the above – antibiotic use and sexual risk behaviour (condom use) - into account in addition to baseline alpha diversity. As above, for menstrual cycle we had a high level of missing data and only 7 women reported intravaginal product use therefore these variables were not included in the final model.

4. In line 149, the authors conduct a multivariate analysis but only include baseline alpha diversity in their model.

We have now included a model that takes additional individual-level parameters into account.

5. Table 3 does not present a multivariate model with confounders to assess changes in taxa. They simply evaluate differentially abundant taxa between and among women.

We have now included a model that takes individual-level confounders – antibiotic use and sexual risk behaviour (condom use) - into account for the cross-sectional analysis. Table 3A shows these data.

For the within participant analysis (pre versus post) the analysis is matched thus naturally taking the individual-level confounders into account (Table 3B).

Overall, while the manuscript is improved to present paired pre-post statistics and figures, the longitudinal modeling could be more rigorous. The authors have an opportunity to set the standard for not only strong study design (randomized and cross-over study) but also they can present robust statistical modeling and assessments. The authors should expand the longitudinal modeling for all outcomes (microbiota and cytokine data).

Thank you for the suggestion on how to strengthen our manuscript. We have included several models that take individual-level confounders into account for the microbiota data and included an adjusted model for the cytokine data as well (Supplementary Table 2B). We believe it is stronger and more streamlined after addressing the comments.

Reviewer #5 (Remarks to the Author):

The manuscript has addressed almost all of the reviewers concerns.

The authors still have not addressed spelling out of abbreviations on first use (Net-EN for example on line 77, COC on line 69, CCVR line 78, CSTs line 104, sPLSD line 260 etc.

This have been corrected for Net-En, COC, CCVR, CST, sPLSDA, sCD40L and IL.

Figure 1 needs further formatting so that the boxes do not cover the text. Similarly Supplemental figure 3 also has formatting issues (but this may be the electronic version)

The boxes do not cover the text in the word doc or the pdf – so this may be an issue with the electronic version received?

Reviewers' comments:

Reviewer #4 (Remarks to the Author):

A randomized cross-over study of hormonal contraception and its effect on microbiota and host immune responses of adolescent females is an incredibly interesting and critical contribution to the literature. At issue is that the authors conducted their sequencing and bioinformatic presentation based on older methods (V4, little refinement of CST IV, taxonomic assignment) and there aren't statistical transition models to confirm the community state types transitions are significant. The multivariable modeling are not rigorous. This paper will set the standard for how we evaluate interventions in the context of the vaginal microenvironment and it is critical that they present meticulous sequencing and statistical methodologies. While the revision is significantly improved, a number of issues remain listed below:

1) Line 74, there is no reference.

2) Line 76, the authors use the term H2O2-producing lactobacilli a number of times and base it on reference 22 (Miller et al OB GYN 2000). The use of the qualifier "hydrogen-peroxide producing" in relation to lactobacilli was recently discussed by Tachedjian et al in Microbiome journal 2018. I refer the author to this paper to consider use of lactic acid producer qualifiers in place of H2O2. This is an important point because readers are still drawn incorrectly to the phenotype of H2O2.

3) Line 86: I think for the uninitiated reader, it is easier to understand phrasing that adolescents assigned to COC had more Lactobacilli (instead of "significantly less diversity").

4) I have concerns about the V4 16S rRNA gene sequencing which clearly can't be re-sequenced at this point. Primers spanning the V3/V4 region identify a greater number of taxa in the vaginal microbiota than V4 alone. Publication in a high profile journal sets a standard for others conducting similar work. I think it is very important that the authors present the issues with V4 sequencing and the resulting taxonomic assignments in a limitations section.

5) Figure 2A is quite difficult to read. Taxonomic assignments are difficult to understand. Better colors need to be used as many species are indistinguishable. The hierarchical clustering is not displayed and it is unclear what the dominating taxa are in CST IV. Why does *L. crispatus* have a *_acidophilus* qualifier? Same for species level resolution at *L. johnsonii_gasseri_taiwanensis*? Presumably the *L_crispatus_acidophilus* is *L. crispatus*. In the text the authors state that

Lachnovaginosum (BVAB1) is the most common taxa in CST IV, however, I don't see it in the taxa list for Figure 2A?

6) Figure 2A is based on Ravel's 2011 paper which cited 5 community state types, however, more recent iterations that the Ravel group has published provide more granular categorization for this important CST IV cluster (IV-A, IV-B, IV-C). See Ravel's more recent work in Nat Commun 2019 and Sci Transl Med. 2012. Ravel group now categorizes low-Lactobacillus state that are driven by various anaerobes in IV-A (ie/BVAB1), with Gardnerella vaginalis and Atopobium vaginae dominating in IVB. More recently a Streptococcus-dominated group has been clustered (IVC). In any case, clustering is a relatively subjective topic but the point remains that clustering to just CST IV is still quite a high level and lacks some resolution. This manuscript could have more distinctly qualified the assembly in IV. If sample size is the issue, the authors can qualify that in limitations.

7) Figure 2B, 2C, 2D: This diagram is informative for displaying the proportion transitioning in CSTs but the diagram is difficult to read (CST I, III and IV aren't labeled) and there are no statistics marking these transitions as statistically significantly different from each other between arms. There is no transition analysis.

8) Also in Figure 2, the only significant within woman change is in COC versus Net-En. I do like these figures showing the pre-post individual data (grey lines).

9) Table 1 shows little/no transactional sex but Table 2 says 97-100% engaged in transactional sex? Also in Table 2, few reported new partners and the line for number of multiple sex partners is blank. Something is wrong here with the data.

10) Additional table 2 comments:

a. Why was pH assessed at a binary 4.7 cut off when 4.5 is generally used for Amsel criteria?

b. I would suggest adding median within woman change in Shannon index. Aggregate number listed isn't that informative. I would want to know how individual women changed.

c. The modeling of menstrual cycle is quite unsatisfactory. The authors acknowledged that adolescents do not report menstrual cycle data reliably. However, to say there is no statistical

difference in menstrual cycle based on median days since last menstrual cycle (21 vs 51 vs 27) is odd. The women on Net-En will likely have more affected bleeding cycles than COC and CCVR. Perhaps modeling bleeding on the day of sampling would be more sufficient because we know that during menstrual bleeding, the vaginal microbiota tend toward CST IV.

d. Did the authors have any comment on why CT and NG were so much higher in CCVR at cross-over?

11) Line 140: While the distribution of CSTs was different at the cross-over visit, based on chi2 test, it is not precise to say CST III was “significantly more prevalent” in COC versus Net-En. The CHI2 test does not assess *L. iners* prevalence specifically.

12) Lines 144-149: Regarding CST transitions from baseline to cross-over, the authors present that there was an increase in CST III and a decrease in CST IV in the COC arm, an increase in CST I and IV in the Net-En arm and an increase in CST IV and III in CCVR. The issue I have with this statement is there is no formal transition modeling or statistical tests to qualify these statements. How do we know these within woman transitions were actually statistically different between arms?

13) In this revised copy of the manuscript, the authors did make additional attempts at multivariable modeling with important co-factors, acknowledging that this was not a blinded randomized trial and therefore participants may change their sexual practices based on which contraceptive they were prescribed. Important variables like condom use might be affected. In this revision, the authors control for baseline alpha diversity, antibiotic use and condom use. Did the antibiotic use also include treatment for symptomatic BV? The authors imply that the antibiotic use was collinear with STI treatment, but what about BV? Was there much BV treatment? Overall, I find the multivariable modeling to still be underwhelming. It is entirely possible that CST need to be used as an interaction term at baseline (instead of controlling for baseline diversity) because the effect of HC may vary by what CST a woman presents with. It is also possible that baseline alpha diversity is in the causal pathway and controlling for it is the incorrect approach to assess the outcomes.

14) One conundrum I have always had with HC interventional studies on the vaginal microbiota, is, for example, can you convert to a *Lactobacillus-crispatus* dominated state at follow-up after HC intervention if you don't have that species present in your microbiota at baseline? Is it presumed that such lactobacilli abundance that it is undetectable --- ie/ that all women have at least one species of Lactobacilli in really low abundance even when they are in a BV state? This would necessitate evaluating interventions by entry microbiota as an interaction term, I think (essentially in stratification).

15) For the DESeq2 analysis (lines 202-210), I would focus on lines 211-214 where the within woman changes in taxa are presented. The aggregate numbers at each visit are not that informative because it depends on where a woman starts as to where she ends. Same for taxa-level analysis on read counts. Within woman change in taxa is what is key.

16) Again, there is a lot of confusion overall in the text as to whether a given result is between arms or within woman pre-post. I think the latter is far more important.

17) Line 320: the authors conclude that Net-En users experienced a greater shift from CST III to I (L. iners to L. crispatus) however there is no statistic for this. We need to be careful about over interpreting descriptive figures.

18) Data availability: only 16S rRNA gene amplicon data are available for public release? What about meta data (questionnaire data) and cytokine data?

Reviewer #4 (Remarks to the Author):

A randomized cross-over study of hormonal contraception and its effect on microbiota and host immune responses of adolescent females is an incredibly interesting and critical contribution to the literature.

At issue is that the authors conducted their sequencing and bioinformatic presentation based on older methods (V4, little refinement of CST IV, taxonomic assignment) and there aren't statistical transition models to confirm the community state types transitions are significant. The multivariable modeling are not rigorous. This paper will set the standard for how we evaluate interventions in the context of the vaginal microenvironment and it is critical that they present meticulous sequencing and statistical methodologies.

In response to these statements, we have consulted a statistician with experience in randomised clinical trials, Dr Zoe Moodie, who has been added to the author list and has assisted us with addressing all comments regarding the statistics.

While the revision is significantly improved, a number of issues remain listed below:

1) Line 74, there is no reference.

- Reference inserted.

2) Line 76, the authors use the term H2O2-producing lactobacilli a number of times and base it on reference 22 (Miller et al OB GYN 2000). The use of the qualifier "hydrogen-peroxide producing" in relation to lactobacilli was recently discussed by Tachedjian et al in Microbiome journal 2018. I refer the author to this paper to consider use of lactic acid producer qualifiers in place of H2O2. This is an important point because readers are still drawn incorrectly to the phenotype of H2O2.

-H2O2 has been replaced with lactic acid.

3) Line 86: I think for the uninitiated reader, it is easier to understand phrasing that adolescents assigned to COC had more Lactobacilli (instead of "significantly less diversity").

-The sentence has been rephrased to say "had more lactobacilli" instead of significantly less diversity.

4) I have concerns about the V4 16S rRNA gene sequencing which clearly can't be re-sequenced at this point. Primers spanning the V3/V4 region identify a greater number of taxa in the vaginal microbiota than V4 alone. Publication in a high profile journal sets a standard for others conducting similar work. I think it is very important that the authors present the issues with V4 sequencing and the resulting taxonomic assignments in a limitations section.

- *The limitations of the V4 region and the taxonomic assignments have now been presented in the Discussion.*

5) Figure 2A is quite difficult to read. Taxonomic assignments are difficult to understand. Better colors need to be used as many species are indistinguishable. The hierarchical clustering is not displayed and it is unclear what the dominating taxa are in CST IV. Why does *L.crispatus* have a *_acidophilus* qualifier? Same for species level resolution at *L.johnsonii_gasseri_taiwanesis*? Presumably the *L_crispatus_acidophilus* is *L.crispatus*. In the text the authors state that *Lachnovaginosum* (BVAB1) is the most common taxa in CST IV, however, I don't see it in the taxa list for Figure 2A?

- *The colors in Figure 2A have been changed to better distinguish the taxa.*
- *A heatmap showing the hierarchical clustering has been added as a supplementary figure (Suppl. Figure 1A).*
- *The qualifiers have been described in the methods section and we refer to a previous paper also describing this in detail. These qualifiers are due to the limitations in regards to the taxonomic annotation which is now described in the discussion.*
- *Lachnovaginosum (BVAB1) - was S.BVAB1 in the taxa list – have been replaced with “Lachnovaginosum (BVAB1)”.*

6) Figure 2A is based on Ravel's 2011 paper which cited 5 community state types, however, more recent iterations that the Ravel group has published provide more granular categorization for this important CST IV cluster (IV-A, IV-B, IV-C). See Ravel's more recent work in *Nat Commun* 2019 and *Sci Transl Med*. 2012. Ravel group now categorizes low-Lactobacillus state that are driven by various anaerobes in IV-A (ie/BVAB1), with *Gardnerella vaginalis* and *Atopobium vaginae* dominating in IVB. More recently a *Streptococcus*-dominated group has been clustered (IVC). In any case, clustering is a relatively subjective topic but the point remains that clustering to just CST IV is still quite a high level and lacks some resolution. This manuscript could have more distinctly qualified the assembly in IV. If sample size is the issue, the authors can qualify that in limitations.

- *Regardless of which clustering method we used, $k=3$ was always the optimal k (i.e. number of clusters), even when we used Ravel et al's method (Jensen-Shannon with Ward D2 linkage). This has been added to the methods.*
*If we force the clustering to use $k=4$, this resulted in a separation of CST-IV into a *Lachnovaginosum* (BVAB1) dominated and a *Gardnerella*/mixed CST-IV subgroup. However, this results in sample sizes too small to perform meaningful statistics.*

7) Figure 2B, 2C, 2D: This diagram is informative for displaying the proportion transitioning in CSTs but the diagram is difficult to read (CST I, III and IV aren't labeled) and there are no statistics marking these transitions as statistically significantly different from each other between arms. There is no transition analysis.

- *The diagrams are clearly labelled with CSTs.*
- *Supplementary Table 1 has been added to provide the number of transitions of CSTs from baseline to crossover within the three study arms. P-values are also provided from omnibus symmetry exact tests for paired contingency tables and included in Supplementary Table 1. There were no statistically significant changes, apart from a trend seen in the Net-En arm. This has now been explained in the manuscript on page 7.*

8) Also in Figure 2, the only significant within woman change is in COC versus Net-En. I do like these figures showing the pre-post individual data (grey lines).

- *Correct and thank you.*

9) Table 1 shows little/no transactional sex but Table 2 says 97-100% engaged in transactional sex? Also in Table 2, few reported new partners and the line for number of multiple sex partners is blank. Something is wrong here with the data.

- *Thank you for noticing; a line was missing and shifted the Table, this has been corrected.*

10) Additional table 2 comments:

a. Why was pH assessed at a binary 4.7 cut off when 4.5 is generally used for Amsel criteria?

- *This has been re-calculated with 4.5 as the cut-off and changed in both tables.*

b. I would suggest adding median within woman change in Shannon index. Aggregate number listed isn't that informative. I would want to know how individual women changed.

- *This has been included in Table 2.*

c. The modeling of menstrual cycle is quite unsatisfactory. The authors acknowledged that adolescents do not report menstrual cycle data reliably. However, to say there is no statistical difference in menstrual cycle based on median days since last menstrual cycle (21 vs 51 vs 27) is odd. The women on Net-En will likely have more affected bleeding cycles than COC and CCVR. Perhaps modeling bleeding on the day of sampling would be more sufficient because we know that during menstrual bleeding, the vaginal microbiota tend toward CST IV.

- *We have removed the statement that there was no statistically significant difference in days since last menstrual period. No samples were taken during menses. Women who were menstruating at the time of their visit had their visit rescheduled. This is spelled out now in the methods.*

- However, in response to the reviewer's request to include menstrual cycle and sexual behaviour changes into the model, we have consulted a statistician. Changes due to randomised intervention are considered more mediators than confounders, and therefore are not usually included in a regression model. We therefore instead performed mediation analyses to determine whether condom use and menstrual patterns after randomization were indeed mediators of the effects. These were very interesting, and show that, in fact, condom use and menstrual cycle provide minimal explanation of the effect of contraception on the vaginal microbiota. This is now shown in supplementary Table 2 and discussed in the text on page 7.

d. Did the authors have any comment on why CT and NG were so much higher in CCVR at cross-over?

- We do find that interesting and have examined this more closely in a separate manuscript.

11) Line 140: While the distribution of CSTs was different at the cross-over visit, based on chi2 test, it is not precise to say CST III was "significantly more prevalent" in COC versus Net-En. The CHI2 test does not assess *L. iners* prevalence specifically.

- We have made it clearer that we were not stating the *L. iners* is more prevalent, only that CST-III is.

12) Lines 144-149: Regarding CST transitions from baseline to cross-over, the authors present that there was an increase in CST III and a decrease in CST IV in the COC arm, an increase in CST I and IV in the Net-En arm and an increase in CST IV and III in CCVR. The issue I have with this statement is there is no formal transition modeling or statistical tests to qualify these statements. How do we know these within woman transitions were actually statistically different between arms?

- P-values to assess differences in the transitions of CSTs from baseline to crossover within the three study arms were calculated using omnibus symmetry exact tests for paired contingency tables and are now included in Supplementary Table 1 and discussed in the text on pgs 6-7.

13) In this revised copy of the manuscript, the authors did make additional attempts at multivariable modeling with important co-factors, acknowledging that this was not a blinded randomized trial and therefore participants may change their sexual practices based on which contraceptive they were prescribed. Important variables like condom use might be affected. In this revision, the authors control for baseline alpha diversity, antibiotic use and condom use. Did the antibiotic use also include treatment for symptomatic BV? The authors imply that the antibiotic use was collinear with STI treatment, but what about BV? Was there much BV treatment? Overall, I find the multivariable modeling to still be underwhelming. It is entirely possible that CST need to be used as an interaction term at baseline (instead of controlling for baseline diversity) because the effect of HC may vary by what CST a

woman presents with. It is also possible that baseline alpha diversity is in the causal pathway and controlling for it is the incorrect approach to assess the outcomes.

- Only symptomatic BV was treated with antibiotics as per standard of care since symptoms of BV or STIs was an exclusion criterion, no participant had BV symptoms at screening. Furthermore, only two participants presented with symptoms between screening and crossover, therefore, there was very limited treatment for symptomatic BV. Therefore, having a STI was truly collinear with antibiotic use.
- When adjusting the linear regression model with baseline CST instead of baseline alpha diversity, the difference in alpha diversity between arms remained significant at crossover (COC vs. CCVR, $p=0.015$ and COC vs. Net-En, $p=0.007$).
- When adjusting the PERMANOVA analysis for baseline CST, we still found significant differences in beta diversity between the three assigned study arms at crossover ($p=0.023$, $R^2=0.042$). Pairwise comparisons revealed that this difference was driven primarily by the differences between CCVR and Net-En ($p=0.008$, $R^2=0.048$). The differences remained significantly different after adjusting for antibiotic use ($p=0.027$, $R^2=0.040$) again driven by differences between the CCVR and Net-En arm ($p=0.011$, $R^2=0.0348$).
- We have included models both with and without adjusting for baseline microbiota (CST) and we have included baseline CST as an interaction term, albeit this was not found to be significant.

14) One conundrum I have always had with HC interventional studies on the vaginal microbiota, is, for example, can you convert to a *Lactobacillus-crispatus* dominated state at follow-up after HC intervention if you don't have that species present in your microbiota at baseline? Is it presumed that such lactobacilli abundance that it is undetectable --- ie/ that all women have at least one species of Lactobacilli in really low abundance even when they are in a BV state? This would necessitate evaluating interventions by entry microbiota as an interaction term, I think (essentially in stratification).

- In the CST-IV group at baseline ($n=64$) (Figure 2A), the median relative abundance of *Lactobacillus* spp. was 0.031 (IQR 0.004-0.144) with none having zero (65% participants had >1% *Lactobacillus*). In terms of specific *Lactobacillus* species, the median relative abundance for *L. crispatus* was 0.0002 (0.00009-0.0007) with 4 having 0. For *L. iners*, the median relative abundance was 0.0207 (IQR 0.002-0.0938) with non having zero (61% participants had >1% *L. iners*). Therefore, we believe that there is always *Lactobacillus* present, albeit sometimes at very low abundance.
- However, adjusting for baseline CST and including baseline CST as an interaction term in our models also takes this into account.

15) For the DESeq2 analysis (lines 202-210), I would focus on lines 211-214 where the within woman changes in taxa are presented. The aggregate numbers at each visit are not that informative because it depends on where a

woman starts as to where she ends. Same for taxa-level analysis on read counts. Within woman change in taxa is what is key.

- *This pre-post (within woman) analysis has been emphasised.*
- *A paired analysis of bacterial read counts for participants changing between methods have now been included (in Suppl. Figure 2A-C).*

16) Again, there is a lot of confusion overall in the text as to whether a given result is between arms or within woman pre-post. I think the latter is far more important.

- *We have clarified whether results are between arms cross-sectionally or pre-post (longitudinal) and emphasised the pre-post results.*
- *A paired analysis of bacterial read counts for participants changing between methods have now been included (in Suppl. Figure 2A-C).*

17) Line 320: the authors conclude that Net-En users experienced a greater shift from CST III to I (*L. iners* to *L. crispatus*) however there is no statistic for this. We need to be careful about over interpreting descriptive figures.

- *A paired analysis of bacterial read counts for participants changing between methods have been included (in Suppl. Figure 2A-C).*
- *Supplementary Table 1 has been added to provide the number of transitions of CSTs from baseline to crossover within the three study arms. P-values are also provided from omnibus symmetry exact tests for paired contingency tables and included in Supplementary Table 1. There were no statistically significant changes, apart from a trend seen in the Net-En arm with the *L.iners* dominant community changing to the *L. crispatus* dominant and the diverse CST. This has now been explained in the manuscript on page 7 and we have been more careful about our interpretation.*

18) Data availability: only 16S rRNA gene amplicon data are available for public release? What about meta data (questionnaire data) and cytokine data?

- *The metadata and cytokine data is available in the same source data file.*